# AUTOCODE: LLMS AS PROBLEM SETTERS FOR COMPETITIVE PROGRAMMING

**Shang Zhou**[1,*]  **Zihan Zheng**[2,*]  **Kaiyuan Liu**[3,*]  **Zeyu Shen**[4,*]  **Zerui Cheng**[4,*]
Zexing Chen[1]  Hansen He[5]  Jianzhu Yao[4]  Huanzhi Mao[7]  Qiuyang Mang[7]
Tianfu Fu[6]  Beichen Li[8]  Dongruixuan Li[9]  Wenhao Chai[4,†]  Zhuang Liu[4,†]
Aleksandra Korolova[4,†]  Peter Henderson[4,†]  Natasha Jaques[3,†]  Pramod Viswanath[4,10,†]
Saining Xie[2,†]  Jingbo Shang[1,†]

[1]University of California San Diego   [2]New York University   [3]University of Washington
[4]Princeton University   [5]Canyon Crest Academy   [6]OpenAI
[7]University of California Berkeley   [8]MIT   [9]University of Waterloo   [10]Sentient Labs

[*]Equal contribution     [†]Advising authors

## ABSTRACT

**Abstract:** Writing competitive programming problems is exacting. Authors must: set constraints, input distributions, and edge cases that rule out shortcuts; target specific algorithms (e.g., max-flow, dynamic programming, data structures); and calibrate complexity beyond the reach of most competitors. We argue that this makes for an ideal test of general large language model capabilities and study whether they can do this reliably. We introduce AutoCode, which uses multiple rounds of validation to yield competition-grade problem statements and test cases. On held-out problems, AutoCode test suites approach 99% consistency with official judgments, a significant improvement over current state-of-the-art methods like HardTests, which achieve less than 81%. Furthermore, starting with a random seed problem, AutoCode can create novel variants with reference and brute-force solutions. By cross-verifying these generated solutions against test cases, we can further filter out malformed problems. Our system ensures high correctness, as verified by human experts. AutoCode successfully produces novel problems judged by Grandmaster-level (top 0.3%) competitive programmers to be of contest quality.

## 1   INTRODUCTION

As Einstein and Infeld put it, "The formulation of a problem is often more essential than its solution, which may be merely a matter of mathematical or experimental skill. To raise new questions, new possibilities, to regard old problems from a new angle requires creative imagination and marks real advances in science." (Einstein & Infeld, 1938). As large language models (LLMs) march toward general-purpose capabilities, with the ultimate goal of artificial general intelligence (AGI), we argue that testing *problem generation* abilities is just as important as *problem solving* abilities. This is particularly true when applying LLMs to advanced programming tasks, where future advancement and economic integration of LLM coding capabilities will require significant validation.

First, problem setting for competitive coding requires a deeper understanding of algorithms that problem solving may not. For example, basic problems can collapse into recognizable templates that can be solved with simple tricks; and many standard programming questions often allow for partial credit or boilerplate solutions that can mask incorrect reasoning. Competitive programming problems have a strict bar designed to assess a deeper understanding of the underlying algorithm design principles, data structures, and complexity trade-offs. Verifying the vast space of possible solutions, along with sufficient coverage of short-cuts or corner cases is challenging, but a necessity for competition programming problems. As such, problem setting encompasses all the challenges of solving a problem, and then more.

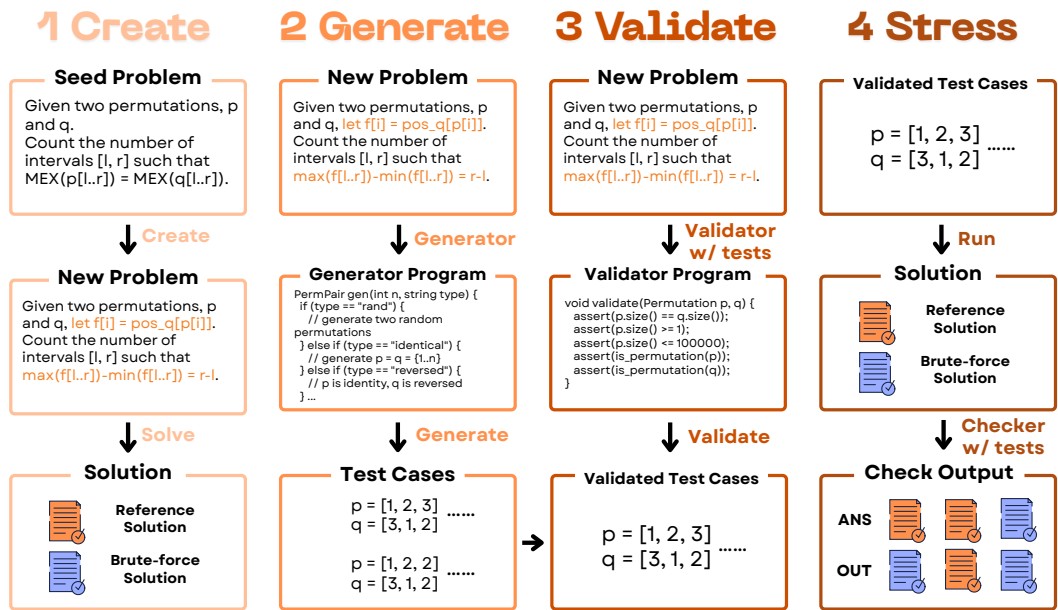

Figure 1: **AutoCode** introduces a closed-loop multi-role Validator-Generator-Checker framework that enables robust test case generation and scalable, self-verified problem generation for competitive programming. It achieves 98.7% consistency with official judgments. This framework precisely mirrors the process human experts follow when creating programming contest problems.

Second, better problem setting will lead to more rigorous competitive programming benchmarks. Since official test data from premier platforms such as Codeforces and AtCoder are not publicly available, researchers currently rely on synthesized datasets such as CodeContests+ (Wang et al., 2025), TACO (Li et al., 2023), and HardTests (He et al., 2025). Our analysis (§4), however, shows that existing test datasets can have both high false positive rates (FPR) and false negative rates (FNR). For instance, a greedy algorithm with poor time complexity might pass a suite of small, random tests, only to fail against adversarially constructed cases designed to expose its flaws. This critical weakness creates a distorted evaluation landscape, rewarding models that discover shortcuts.

Third, successful problem setting of novel challenges may pave the path for self-improving models and AGI, as well as validating deployments in complex software stacks.

LLMs are already used by over a billion people and increasingly for coding applications, saving hundreds of millions of dollars with LLM-assisted programming (OpenAI, 2025; Zeff, 2025; AWS DevOps Blog, 2024), albeit largely on relatively simple tasks. To expand LLM capabilities toward the strategic reasoning and long-term planning that real programming demands, and to integrate LLMs safely within the software development stack, we must be able to determine, with high fidelity, whether model-generated code is a valid solution to the problem at hand. Misspecified rewards, in the form of malformed problem formulation or verification, can cause models to optimize for the wrong thing or fall into bad local optima. We find, for example, that current benchmark datasets suffer from an even higher FNR than FPR, often caused by invalid inputs to unit tests; as a result, a perfectly valid, creative solution can be unjustly penalized when it crashes or produces an incorrect output on malformed input data. This pollutes data for reinforcement learning by punishing valid lines of reasoning, while high FPR fails to penalize flawed ones.

To address these critical gaps, we introduce AutoCode, a systematic framework that employs LLMs in a closed-loop, multi-role system to automate the entire lifecycle of competitive programming problem creation and evaluation. Our first contribution is an enhanced Validator-Generator-Checker framework that achieves state-of-the-art reliability in test case generation, where the Generator generates test cases, the Validator validates whether the generated test cases satisfy the problem constraints, and the Checker checks whether a solution is correct given the test cases. We move beyond standard implementations with several key heuristics: the LLM generates multiple candidate Validators and Checkers, and we select the most robust one through targeted testing. The generator executes a multi-pronged strategy, creating a diverse and adversarial test suite that includes small-data

exhaustion, randomized stress tests, and TLE-inducing cases designed to expose incorrect time complexities. Building on this highly reliable verification pipeline, we introduce our second major contribution: a novel process for generating new, high-quality problems. This process begins with a seed problem to inspire the LLM in a promising direction. To ensure correctness without human intervention, we employ a dual verification protocol: the LLM generates a problem statement, an efficient reference solution, and a simple brute-force solution. The new problem is accepted only after rigorously verifying that the efficient solution's output matches the ground truth established by the brute-force solution across all test cases. In summary, our work makes the following contributions:

- **State-of-the-Art Test Cases Generation for Existing Problems.** Our testcase generation framework achieves over 91% consistency with official judgments on a large-scale benchmark of 7538 problems, significantly outperforming existing methods whose consistency ranges from $72 - 81\%$. We also point out that the ability to generate test cases is not only useful for competitive programming, but also has broad practical significance for general tasks involving input-output matching.

- **Novel and High-Quality Problems Generation.** We pioneer a systematic process that uses a dual-verification mechanism to generate, validate, and score new problems. Vetted by elite competitive programmers, this process produces novel problems deemed high-quality and original enough for official contests, with problems passing automated verification achieving a correctness rate of 94%.

## 2 RELATED WORKS

LLMs show rapid progress in code generation recently (Hui et al., 2024; Deepseek, 2024; Zhu et al., 2024; Gong et al., 2025; Xie et al., 2025). Recent efforts have progressed along three threads. First, new benchmarks have been proposed to target diverse problem types. These range from standard coding interviews (Jain et al., 2024) and competitive programming (Zheng et al., 2025) to challenging domain-specific problems (Beniamini et al., 2025) and human-LLM collaboration scenarios (Yang et al., 2025). Second, a growing line of work focuses on strengthening verdict reliability. For example, (Liu et al., 2023) augments existing unit tests. A more recent trend involves rule- or LLM-driven generators that create new challenging test cases (Li et al., 2023; Wang et al., 2025; He et al., 2025; Cao et al., 2025; Sung et al., 2025; Ma et al., 2025). These frameworks are designed to emphasize constraint validity, edge-case coverage, and adversarial-ness. Third, solver / data-centric approaches have aimed to scale solution search or curate verified corpora. Methods like AlphaCode (Li et al., 2022), AceReason (Chen et al., 2025b), Absolute Zero (Zhao et al., 2025), and rStar-Coder (Liu et al., 2025) leverage signals from reinforcement learning or self-play (Zelikman et al., 2022; 2024; Shinn et al., 2023; Madaan et al., 2023; Chen et al., 2025a). (Huang et al., 2026) manually create test cases for evaluation. However, these methods focus on solving or data curation rather than offering end-to-end testing pipelines (Zhang et al., 2023). In contrast, AutoCode unifies and extends these three directions. It couples a Validator-Generator-Checker (and interactor) loop to enforce test legality and adversarial coverage. Critically, AutoCode also employs a dual-verification protocol (using a reference solution and a brute-force solution) to generate and certify entirely new problems. This framework allows AutoCode to simultaneously address two key challenges in the field: static-benchmark contamination and under-constrained test cases.

## 3 TEST CASE GENERATION

Our test case generation process is a structured framework designed for maximum rigor and coverage. The framework as shown in Figure 1 starts with the Validator, which serves as the cornerstone of the entire system. Its function is to ensure any given input strictly adheres to all constraints specified in the problem description. A Validator is critical for minimizing the FNR, as it prevents correct programs from failing on malformed data. Next, the Generator employs diverse strategies to create a broad spectrum of inputs, aiming to reduce the FPR, where incorrect or inefficient programs are erroneously judged as correct. Any invalid cases produced by the Generator are subsequently filtered by the Validator, ensuring we obtain a high-quality set of inputs. Finally, to assess the contestant's output, a Checker compares it against the reference solution's output, while for interactive tasks, an Interactor engages in a multi-turn dialogue with the contestant's program to issue a final verdict. Terminology is defined in Appendix A. Because one of our prominent goals is to serve as a high-quality verifier for RLVR, we pay particular attention to reducing the FPR. We distinguish test cases (input-answer pairs) from the test data, which includes the Checker and Interactor programs required for evaluation.

## 3.1 VALIDATOR

---

**Algorithm 1:** BUILDVALIDATOR

**Input:** Problem spec $\mathcal{S}$
**Output:** Selected Validator $V^\star$
1 $\mathcal{E} \leftarrow$ LLM.GENERATEEVALCASES$(\mathcal{S};\ N{=}40,\ 10\ \text{valid},\ 30\ \text{near-valid})$
2 $\{V_1, V_2, V_3\} \leftarrow$ LLM.EMITVALIDATORS$(\mathcal{S},\ K{=}3)$
3 **for** $V$ **in** $\{V_1, V_2, V_3\}$**:**
4 $\quad$ score$(V) \leftarrow \sum_{(x,\text{label})\in\mathcal{E}}[V(x){=}\text{label}]$
5 $V^\star \leftarrow \arg\max_{V\in\{V_1,V_2,V_3\}} \text{score}(V)$
6 **return** $V^\star$

---

The foundation of our framework is a robust Validator responsible for rejecting any input that violates the problem's explicit constraints. To construct this Validator, we first prompt the LLM to generate a targeted suite of 40 test cases. This suite consists of 10 valid inputs and 30 "near-valid" invalid inputs, i.e., cases that are almost valid but contain subtle violations (see Appendix D for an example). With this evaluation set, we then prompt the LLM to generate three distinct candidate Validator programs. Each candidate Validator is benchmarked against these 40 cases. The one that correctly classifies the most valid and invalid cases is selected as the winning Validator. This Validator is then used in all subsequent stages to filter malformed inputs.

## 3.2 GENERATOR

---

**Algorithm 2:** BUILDGENERATORSUITE

**Input:** $\mathcal{S}$, Validator $V^\star$
**Output:** Final test set $\mathcal{T}$
1 $\mathcal{G}_1 \leftarrow$ EXHAUSTIVESMALL$(\mathcal{S})$       `# small-scale enumeration coverage`
2 $\mathcal{G}_2 \leftarrow$ RANDOMEXTREME$(\mathcal{S})$   `# random + extreme: overflows, precision,`
  `hash collisions, etc.`
3 $\mathcal{G}_3 \leftarrow$ TLEINDUCING$(\mathcal{S})$       `# worst-case inputs inducing TLE`
4 $\mathcal{U} \leftarrow \mathcal{G}_1 \cup \mathcal{G}_2 \cup \mathcal{G}_3$
5 $\mathcal{U} \leftarrow \{x \in \mathcal{U} \mid V^\star(x){=}\text{valid}\}$       `# Validator filters invalid inputs`
6 $\mathcal{U} \leftarrow$ DEDUPBYSIGNATURE$(\mathcal{U})$   `# hash/normalization-based deduplication`
7 $\mathcal{U} \leftarrow$ BALANCEBUCKETS$(\mathcal{U};\ \text{size/structure/hardness})$
8 $\mathcal{T} \leftarrow$ SAMPLEWITHCOVERAGE$(\mathcal{U};\ \text{target size})$
9 **return** $\mathcal{T}$

---

With a reliable Validator, the Generator's main goal is to maximize test coverage. It is designed to create challenging test cases that identify incorrect or inefficient solutions, which could reduce the false positive rate (FPR). To achieve this, we combine three distinct strategies:

- **Exhaustive Small-Case Coverage.** For problems with small constraints, we exhaustively generate all permutations and combinations of small-scale data. This ensures complete coverage of fundamental boundary conditions.

- **Randomized and Extreme Data.** We generate large-scale random inputs to stress-test solutions. This targets common failure modes such as integer overflows, floating-point precision, and array out-of-bounds. Drawing from contest experience, we also design adversarial "hack" cases, such as those specifically intended to break common greedy algorithms or induce hash collisions.

- **TLE-Inducing Data.** To identify solutions with incorrect time complexity, a common issue where slow algorithms pass weak test suites, we construct inputs with specific structures designed to maximize the running time. This ensures that only solutions that meet the intended time complexity can pass.

---

**Algorithm 3:** BUILDCHECKER

---

**Input:** $\mathcal{S}$, reference solution $\mathcal{R}$, Validator $V^\star$
**Output:** Selected Checker $C^\star$
1   $\mathcal{Q} \leftarrow$ LLM.GENERATECHECKERSCENARIOS$(\mathcal{S}, \mathcal{R};\ N{=}40)$
2   **foreach** $q \in \mathcal{Q}$ **do**
3     **if** $V^\star(q.\texttt{input}) \neq$ **valid:**
4       remove $q$
                                          `# ensure input legality`
5   $\{C_1, C_2, C_3\} \leftarrow$ LLM.EMITCHECKERS$(\mathcal{S},\ K{=}3)$
6   **for** $C$ **in** $\{C_1, C_2, C_3\}$**:**
7     $acc(C) \leftarrow \frac{1}{|\mathcal{Q}|} \sum_{q \in \mathcal{Q}} [\, C(q.\text{input},\ q.\text{contestant\_out},\ q.\text{ref\_out}){=}q.\text{verdict}\,]$
8   $C^\star \leftarrow \arg\max_C acc(C)$
9   **return** $C^\star$

---

### 3.3 CHECKER

The Checker's role is to determine the final verdict (e.g., Accepted or Wrong Answer) by comparing a submission's output against the reference solution's output. To ensure the quality of the Checker, we follow an analogous construction process as the Validator. First, we prompt the LLM to generate 40 distinct test cases. Each test case includes a valid input, a reasonable-looking output that is potentially incorrect, the correct reference output, and the expected verdict. The validity of all inputs is guaranteed by the winning Validator. Next, we prompt the LLM to generate three candidate Checker programs. We evaluate each candidate Checker against the 40 test cases. The checker that achieves the highest accuracy in matching the expected verdicts is selected as the winning Checker.

### 3.4 INTERACTOR

---

**Algorithm 4:** BUILDINTERACTOR

---

**Input:** $\mathcal{S}$, reference solution $\mathcal{R}$, Validator $V^\star$
**Output:** Selected Interactor $I^\star$
1   $\mathcal{M} \leftarrow$ LLM.MUTATE$(\mathcal{R};\ \text{small logical edits: } <>/ \leq$
    $/ \geq$ swap, off-by-one, missing checks, wrong tie-breaks, RNG misuse, etc.)
2   $\{I_1, I_2, I_3\} \leftarrow$ LLM.EMITINTERACTORS$(\mathcal{S},\ K{=}3)$
3   **for** $I$ **in** $\{I_1, I_2, I_3\}$**:**
4     $p \leftarrow [\,$SIMULATE$(I, \mathcal{R}, V^\star){=}$Accepted$\,]$
5     $f \leftarrow \sum_{m \in \mathcal{M}} [\,$SIMULATE$(I, m, V^\star){=}$Rejected$\,]$
6     score$(I) \leftarrow (p,\ f)$     `# lexicographic: prioritize passing the true`
       `solution, then maximize discrimination`
7   $I^\star \leftarrow \arg\max_I$ score$(I)$
8   **return** $I^\star$

---

Our framework introduces a novel, fully automated approach for generating test data for interactive problems, a previously unaddressed challenge (e.g., in CodeContests+ (Wang et al., 2025)). The core innovation lies in a mutant-based discrimination process. We begin by prompting the LLM to perform several small but critical logical modifications to the provided reference solution, thereby creating a set of slightly incorrect mutant programs. Examples are shown in Appendix E. These mutants serve as challenging foils for the Interactor. The LLM then generates three candidate Interactor programs. The crucial selection criterion is identifying the Interactor that most effectively distinguishes the correct, unmodified reference solution from the flawed mutant versions during a simulated interaction shown in Appendix G. The Interactor that successfully passes the reference solution while failing the maximum number of mutants is chosen, proving its ability to robustly probe for the specific logic required by the problem.

Table 1: Performance comparison on the 7538-problem benchmark. The evaluation is performed on a set of 195,988 human submissions randomly taken from the CodeContests dataset, where each problem has 26 submissions, 50% of which are correct and 50% are incorrect. Results are reported with 95% confidence intervals. The results for AutoCode are obtained using `o3`.

| Method | Consistency (%) ($\uparrow$) | FPR (%) ($\downarrow$) | FNR (%) ($\downarrow$) |
|---|---|---|---|
| CodeContests (Li et al., 2022) | $72.9 \pm 0.2$ | $7.7 \pm 0.2$ | $46.3 \pm 0.3$ |
| CodeContests+ (Wang et al., 2025) | $79.9 \pm 0.2$ | $8.6 \pm 0.2$ | $31.6 \pm 0.3$ |
| TACO (Li et al., 2023) | $80.7 \pm 0.2$ | $11.5 \pm 0.2$ | $26.9 \pm 0.3$ |
| HardTests (He et al., 2025) | $81.0 \pm 0.2$ | $12.1 \pm 0.2$ | $25.8 \pm 0.3$ |
| AutoCode (Ours) | $\mathbf{91.1 \pm 0.1}$ | $\mathbf{3.7 \pm 0.1}$ | $\mathbf{14.1 \pm 0.2}$ |

Table 2: Ablation study results on the 720-problem benchmark. For each problem, the evaluation uses 33 submissions generated by different LLMs, 25% of which are accepted. This setup is particularly relevant for reinforcement learning applications. Each row shows performance after removing a single component from the full framework. All results in this table are generated by `GPT-5-High`.

| Configuration | Consistency (%) ($\uparrow$) | FPR (%) ($\downarrow$) | FNR (%) ($\downarrow$) |
|---|---|---|---|
| w/o Generator Strategy 1 (Exhaustive) | $98.4 \pm 0.2$ | $1.7 \pm 0.2$ | $1.3 \pm 0.3$ |
| w/o Generator Strategy 2 (Random/Extreme) | $98.4 \pm 0.2$ | $1.6 \pm 0.2$ | $1.3 \pm 0.3$ |
| w/o Generator Strategy 3 (TLE) | $98.6 \pm 0.2$ | $1.4 \pm 0.2$ | $1.3 \pm 0.3$ |
| w/o Prompt Optimization | $98.0 \pm 0.2$ | $1.8 \pm 0.2$ | $2.9 \pm 0.4$ |
| AutoCode Full Framework | $\mathbf{98.7 \pm 0.1}$ | $\mathbf{1.3 \pm 0.2}$ | $\mathbf{1.2 \pm 0.3}$ |

# 4 BENCHMARKING TEST CASE ROBUSTNESS

To rigorously evaluate our test case generation framework, we establish two distinct benchmarks. The primary benchmark consists of 7538 problems derived from the intersection of well-known existing datasets: CodeContests+ (Wang et al., 2025), CodeContests, HardTests (He et al., 2025), and TACO (Li et al., 2023). Notably, this large-scale set does not contain interactive problems, and due to the filtering inherent in these datasets, its average difficulty for test data generation is slightly lower than a typical Codeforces round.

To address this and test our system under more challenging, real-world conditions, we create a second benchmark of 720 recent, rated problems from Codeforces. This set is completely unfiltered, including notoriously difficult-to-handle interactive problems and those requiring complex, structured test data. We are unable to evaluate prior methods on this newer benchmark as their data generation codebases are not publicly available.

Our evaluation is based on three key metrics. Consistency measures the overall percentage of agreement between the verdicts from our tests and the official judgments. We further dissect disagreements into two critical error rates. The FPR is defined as the proportion of officially incorrect solutions that are erroneously accepted by our generated tests. Conversely, the FNR is the proportion of officially correct solutions that are erroneously rejected by our tests.

**Comparison with other baselines.** We evaluate AutoCode on the benchmark of 7538 problems against four leading baselines. As detailed in Table 1, our framework achieves 91.1% consistency with official judgments. This marks a significant leap over existing methods which don't surpass 81.0%. Critically, AutoCode substantially reduces the FPR to just 3.7% and the FNR to 14.1%, representing $\approx 50\%$ decrease in both metrics over the current state-of-the-art. Figure 2 shows the error verdicts distribution, showing that most of the problems are consistent with ground truth verdicts.

To further test our system's robustness, we curate a more challenging benchmark of 720 recent, unfiltered Codeforces problems, including complex interactive tasks. As shown in Table 2, AutoCode maintains its exceptional performance, achieving 98.7% consistency. This result validates our method's effectiveness on modern, difficult problems where prior methods could not be evaluated.

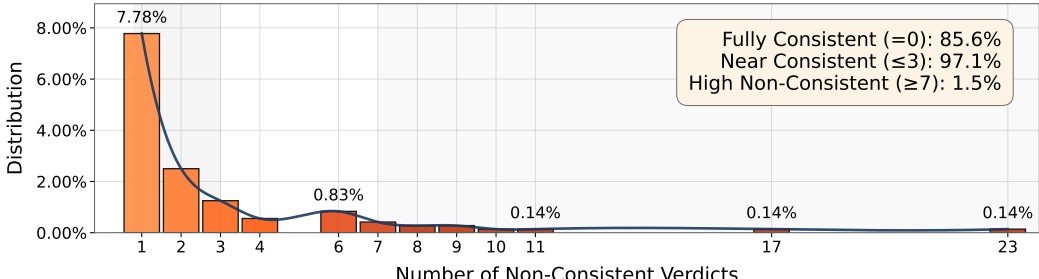

Figure 2: **Distribution of Non-Consistent Verdicts.** Each problem in the 720-problem benchmark is evaluated using 33 submissions generated by different LLMs. The verdicts for 85.6% of the problems are consistent with the official judgments, and 97.1% have three or fewer inconsistencies.

**Ablation studies.** We further conduct an ablation study to determine how each part of AutoCode affects overall performance. The complete AutoCode framework sets a strong baseline, achieving 98.7% consistency with official judgments, along with a 1.3% FPR and a 1.2% FNR. Prompt optimization turns out to be especially important; removing it drops consistency to 98.0% and more than doubles the FNR to 2.9%. The three different generator strategies also significantly complement each other. Removing either the exhaustive or random/extreme strategy raises the FPR to 1.7% and 1.6%, respectively, highlighting their role in catching flawed solutions.

Validator and Checker selection tests are also crucial. These tests pick the most reliable output from several model-generated candidates, helping weaker models avoid logical errors. Removing these tests increases the FNR to 1.3% and 1.4%. A good Validator prevents correct solutions from being rejected due to bad inputs, while a strong Checker accurately evaluates outputs with complex judgment rules. Even small improvements (0.1%) matter, as they help address problems that are very difficult to generate test cases for.

## 5 PROBLEM GENERATION

Our novel problem generation framework builds upon the robust test generation framework described in Section 3 and shown in Figure 1, but introduces a critical dual-verification protocol to ensure correctness without human intervention.

Each generated problem is graded on a 6-level scale, judged by top human competitive programmers. We interviewed 8 human expert problem setters, all of whom report that they often build on a specific existing problem when authoring new problems. By adding, removing, or modifying certain conditions of such a "seed problem," they create new and often more difficult problems that require novel insights. Inspired by their insights, our approach begins by selecting a random Codeforces problem (with difficulty rating less than 2200) as a "seed problem." The LLM is tasked with generating a new problem by adding, deleting, or modifying certain conditions from this seed problem, along with an efficient reference solution (std.cpp) and a brute-force solution (brute.cpp). brute.cpp usually has higher time complexity but is very unlikely to be incorrect, so we leverage it to stress-test the validity of the problem. Using our enhanced test case generation technique, we construct a comprehensive set of test data that fully covers small cases. Both brute.cpp and std.cpp are then executed on this dataset. A problem is only deemed correct if, for every test case, both programs' outputs (where the brute-force solution may legitimately fail to finish due to timeout) are pairwise validated by the checker as consistent answers and outputs. This dual-verification protocol, where brute.cpp serves as the initial ground truth, and the validated reference solution undergoes an additional full test generation cycle, successfully filters out 27% of error-prone problems, raising the correctness rate of LLM-provided reference solutions from 86% to 94%.

After filtering, over 80% of the problems are annotated as having sufficient quality to serve as training data for models, and 23% of the problems have novel or creative designs involved. We present the detailed rubrics and the score distribution in Figure 3. In the following, we summarize several key findings regarding the performance of LLMs in problem generation. We also present the human expert grading criteria in Appendix F and one example under ICPC/IOI-level in Appendix B.

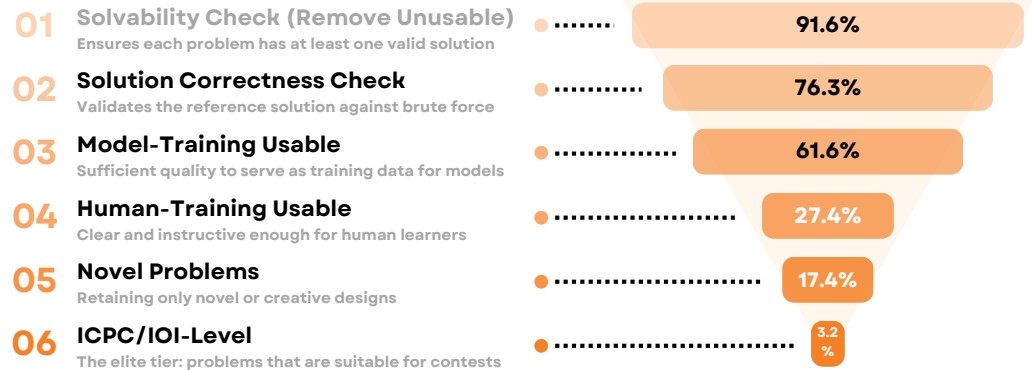

Figure 3: **AutoCode** is capable of automatically generating novel competitive programming problems. After careful examination and filtering by multiple human experts, 61.6% achieve a quality level suitable for large language model training, and 3.2% are considered good enough to serve as problems for ICPC/IOI-level contests. Level 1 and 2 correspond to automatic correctness checks performed by the system, while level 3 through 6 involve human evaluation to differentiate among finer levels. It is noted that only the Level 1 questions are wrong, while the Level 2 – 6 questions are all correct and usable, with the only difference being their quality. Detailed grading rubrics are in Appendix F.

> ***Finding* 1.** LLMs can generate solvable problems that they themselves are unable to solve.

LLMs can generate *solvable* problems that they themselves are *unable to solve*. In our experiments, about 4.2% of the problems fall into this category. In other words, the model is indeed capable of creating logically sound problems that can be solved by other models or by humans, but due to limitations in its own reasoning ability, it fails to solve them correctly. Ideally, these problems could serve as sources for model self-improvement, and we think this is a very interesting phenomenon that is worth further investigation. We provide an example in the Appendix C.

> ***Finding* 2.** LLMs tend to create new problems by combining existing problem frameworks and emphasizing knowledge and implementations.

LLMs tend to concatenate, combine, or embed existing algorithmic knowledge or tricks into established problem frameworks, rather than proposing entirely new problem models that require original solution strategies. This form of recombinational innovation primarily reflects reuse of existing knowledge rather than expansion of creative thinking. Such a tendency highlights a fundamental divergence between humans and LLMs in defining novelty: while humans emphasize originality in modes of reasoning and problem-solving ideas, LLMs rely more heavily on recombination and reconfiguration of pre-existing knowledge. Also, the generated problems often place greater emphasis on assessing specific algorithmic knowledge or demanding complex implementation details, while less often rely on clever design and subtle observation.

> ***Finding* 3.** Novel problems tend to have larger difficulty gain over seed problems, and the generated problems have the highest quality when the corresponding seed problem is moderately difficult.

We observe that the difficulty shifts induced by LLMs during problem adaptation follow a systematic pattern rather than occurring randomly. On average, adapted problems become approximately 334 Elo score harder; those judged as "novel" show an average increase of 498 score, whereas non-novel problems increase by only about 108 score. High-quality problems are predominantly generated when the original seed problem lies in the moderately high difficulty range. For overly difficult seed problems, the LLM has limited room to introduce effective modifications, resulting in minimal difficulty gains and insufficient novelty. Conversely, for overly easy seed problems, even after an average increase of 334 score, the absolute difficulty remains too low to meet high-quality standards. Notably, around 5% of generated problems fall into a critical zone, with pass@1 scores between 0.1 and 0.5, meaning the model sometimes succeeds and sometimes fails. These boundary cases present a valuable opportunity for constructing high-quality self-play datasets, enabling models to enhance their capabilities through repeated attempts at solving such borderline problems.

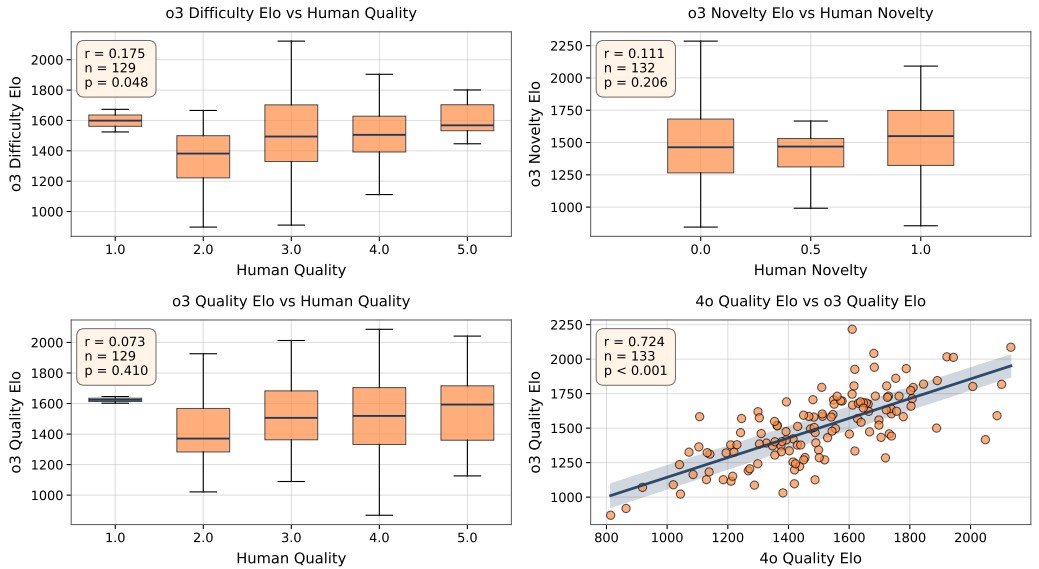

Figure 4: Correlations between human experts and LLM judgments on generated problems. The charts illustrate a significant gap between human and LLM perceptions of quality and novelty (Finding 4), despite high inter-LLM agreement (bottom right). LLM-predicted difficulty shows a weak positive correlation with human-rated quality (top left), suggesting it can serve as a noisy proxy for quality estimation (Finding 5).

> ***Finding 4.*** Human experts and LLMs show almost no correlation in their judgment of problem quality and novelty.

We identify a significant divergence between humans and LLMs in their judgment of problem quality and novelty. To quantify this discrepancy, we employ an Elo Rating scheme to assess LLM judgments, following the methodology described in (Zhou et al., 2024). The correlation coefficients between the `o3` and human experts are only 0.07 for quality and 0.11 for novelty, indicating a substantial misalignment between the model's internal evaluation standards and human expert criteria. Interestingly, both humans and LLMs demonstrate high within-group consistency in terms of quality: correlations among human experts reach 0.71, while `GPT-4o` and `o3` show a correlation of 0.72. These findings suggest that relying solely on LLMs to self-evaluate the quality of their generated problems is inadequate, and more sophisticated evaluation mechanisms are required to align with human preferences as also shown in Figure 4.

> ***Finding 5.*** Difficulty of the generated problem and difficulty gain over the seed problem serve as a better indicator of problem quality than LLM self-evaluations.

While LLMs are unreliable in directly assessing problem quality, leveraging their predictions of problem difficulty provides a more effective indirect measure. First, difficulty is the strongest predictor of quality: correlations between human-assigned quality scores and both absolute problem difficulty and difficulty gain reach as high as 0.60. Second, difficulty gain outperforms novelty as an indicator. As a continuous variable, it captures richer information, since problems that exhibit sufficiently large difficulty increases tend to incorporate new ideas or more complex combinations, thereby manifesting novelty. Finally, LLMs possess a certain ability to estimate problem difficulty; by exploiting this capacity, difficulty can be used as a proxy signal to indirectly predict problem quality, achieving correlations of around 0.18 as shown in Figure 4.

## 6    CONCLUSION

In this work, we introduce AutoCode, a closed-loop multi-role framework that leverages LLMs as problem setters for competitive programming. By coupling a Validator-Generator-Checker (and Interactor) framework with a dual-verification protocol, AutoCode achieves state-of-the-art reliability in test case generation and extends beyond prior methods to generate entirely new, contest-quality

problems. Extensive experiments on over 7,500 problems and recent Codeforces benchmarks demonstrate that AutoCode substantially reduces both false positives and false negatives, achieving more than 98% agreement with official judgments and successfully producing novel problems validated by expert programmers. Beyond test generation, our analysis sheds light on both the strengths and weaknesses of LLMs in creative problem authoring. While models excel at recombination of algorithmic knowledge, they struggle to introduce truly novel reasoning paradigms or flawless sample design. Nevertheless, we show that difficulty and difficulty gain serve as reliable proxy signals for problem quality, providing a scalable path toward self-play.

## ETHICS STATEMENT

This study adheres to the ethical standards of academic research. The benchmark datasets we used are all publicly available from competitive programming platforms (such as Codeforces and AtCoder), and do not involve any personal privacy data or sensitive information. All data were filtered and anonymized, and used solely for algorithm evaluation and model validation. Furthermore, the experiments did not cause any direct or indirect harm to individuals or groups. We recognize that automated problem generation may be misused, for example in cheating or spreading low-quality content. Therefore, we emphasize the critical role of human experts in the final screening and evaluation. We advocate for the responsible and reasonable use of the outcomes of this research.

## REPRODUCIBILITY STATEMENT

To ensure reproducibility, we provide a complete description of the framework design, including the construction of the Validator, Generator, Checker, and Interactor (see Section 2 of the main text). Algorithm pseudocode and process diagrams are detailed in the appendix, along with concrete steps for validation data construction, problem generation, and the dual-verification protocol. All experiments were conducted on publicly available benchmarks (CodeContests+, TACO, HardTests, etc.) as well as on recent Codeforces problems. We report ablation studies (Table 2) and comparisons with existing methods (Table 1), supplemented with statistical significance analysis.

## THE USE OF LARGE LANGUAGE MODELS

The main contributions of this work rely on the generative and discriminative capabilities of LLMs. Specifically, LLMs are employed in a multi-role framework as generator, validator, and checker. During training and experimentation, we use publicly accessible LLM APIs (*e.g.,* `GPT-5-High`) without additional fine-tuning. It is important to note that while LLMs can automatically generate new problems, the quality and novelty of these problems still require expert evaluation and filtering. We firmly believe LLMs should serve as auxiliary tools to expand data generation and test coverage, rather than as replacements for human creative judgment. In addition, LLMs are partially used during the writing stage of this paper for language polishing and improving clarity of expression.

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

# Appendix

The appendix is structured as follows:

- Preliminaries and definitions of key concepts and terms in Section A.
- Example of Competition-Grade level generated problem in Section B.
- Example of generated problem that LLM itself can't solve in Section C.
- Example of valid inputs and near-valid illegal inputs in Section D.
- Example of incorrect mutant programs in Section E.
- Human experts grading policy on LLM generated problems in Section F.
- Pseudo code of simulator in interactor in Section G.
- Limitations and future work in Section H.
- Prompt templates in Section I.
- Early results of RL on AutoCode data in Section J.
- Fair comparison with Hardtests in Section K.
- Human evaluation process and requirements in Section L.

## A  PRELIMINARIES

In this section, we provide background information on the judgment process in competitive programming, define key terminologies used throughout the paper, and detail the core components of the AutoCode framework.

### A.1  COMPETITIVE PROGRAMMING SOLUTION JUDGMENT

In competitive programming, solutions are judged by executing them against a series of test cases. Each test case consists of an input and a ground-truth output. The system compares the program's output for each input against the ground-truth answer. A solution is only considered correct if it passes all test cases while executing within the problem's time and memory constraints. The outcome of the judgment process is a single verdict. The most common verdicts are:

- **Accepted (AC):** The solution passed all test cases within the given time and memory limits. This is the only successful outcome.
- **Wrong Answer (WA):** The solution produces outputs that fail on at least one test case.
- **Time Limit Exceeded (TLE):** The solution exceeds the time limit on at least one test case.
- **Memory Limit Exceeded (MLE):** The solution exceeds the memory limit on at least one test case.
- **Runtime Error (RE):** The solution terminates abnormally due to an error like a segmentation fault or division by zero.

### A.2  KEY TERMINOLOGIES

**Codeforces.**  A popular online platform that hosts competitive programming contests. The difficulty of problems on this platform is often used as a benchmark.

**Elo rating for problem difficulty.**  On Codeforces, each problem has a numerical rating assigned to a problem to quantify its difficulty. It's calibrated based on the performance of contestants on this problem during the competition. In general, a difficulty rating of $\leq 2000$ is deemed as easy to medium range, $(2000, 3000]$ is deemed as medium to hard range, and $\geq 3000$ is deemed as very hard (Zheng et al., 2025).

**Generator.**  A program that produces the input files for the test cases. A robust set of generators is needed to create a diverse set of test cases that cover edge cases, average cases, and worst-case scenarios designed to challenge the time or memory complexity of incorrect algorithms.

**Validator.** A program that verifies whether a test case conforms to the constraints described in the problem statement (e.g., $1 \leq N \leq 1000$, the given graph is a tree).

**Checker.** A program that compares a contestant's output to the correct answer and issuing a verdict. Many problems require custom Checkers beyond simple text comparisons, including

- **Problems with multiple correct solutions**: For example, in a constructive problem asking for any valid path in a graph, the Checker must verify the validity of the contestant's proposed path, not just match it to one specific example.
- **Problems with floating-point outputs**: When the answers are floating-point numbers, answers are accepted if they are within a certain absolute or relative error tolerance (e.g., $10^{-6}$).
- **Problems where output format is flexible**: For instance, some problems that require contestants to output "Yes" or "No" may also accept "YES" or "NO" as correct answers.

**Interactor.** A specialized component is used for interactive problems. In an interactive problem, a contestant's solution does not receive all input at once. Instead, it engages in a real-time dialogue with the judge's program (the Interactor). The solution makes a series of queries to the Interactor, receives responses, and use the information gathered from this dialogue to determine the final answer. A classic example is a guessing game where the Interactor knows a secret number, and the solution must find it by making queries and receiving "higher" or "lower" as responses.

# B  EXAMPLE OF LLM GENERATED PROBLEM: ROW–COLUMN PORTAL

**Time limit per test:** 2 seconds
**Memory limit per test:** 256 megabytes

You are given a binary grid $A$ with $n$ rows and $m$ columns ($0 = $ empty, $1 = $ obsidian).

Fix two integers $a$ and $b$ with $5 \leq a \leq n$ and $4 \leq b \leq m$. A sub-rectangle $M$ of size $a \times b$ (with rows $r..r + a - 1$ and columns $c..c + b - 1$) is called a **portal** if and only if:

- Every cell on the border of $M$ is 1 **except** the four corner cells (which can be either 0 or 1).
- Every strictly interior cell of $M$ is 0.

You are allowed to choose **one** sub-rectangle of size $a \times b$ and then perform the following operation any number of times **inside that chosen sub-rectangle only**:

- Pick a row of the sub-rectangle and flip all its cells ($0 \leftrightarrow 1$), or
- Pick a column of the sub-rectangle and flip all its cells ($0 \leftrightarrow 1$).

Flips affect only the chosen $a \times b$ sub-rectangle (not the rest of the grid). Your task is to find the **minimum number of flips** (rows + columns) needed to turn **some** $a \times b$ sub-rectangle into a portal. If it is impossible for every placement, print $-1$.

### INPUT

The first line contains an integer $t$ ($1 \leq t \leq 2 \cdot 10^5$) — the number of test cases.

For each test case:

- The first line contains four integers $n, m, a, b$ ($5 \leq a \leq n \leq 2000$, $4 \leq b \leq m \leq 2000$).
- Then follow $n$ lines, each a string of length $m$ consisting of characters 0 and 1.

It is guaranteed that the sum of $n \cdot m$ over all test cases does not exceed $4 \cdot 10^5$.

### OUTPUT

For each test case, output a single integer — the minimal number of flips needed, or $-1$ if impossible.

EXAMPLES

**Input**

```
1
5 4 5 4
1111
1001
1001
1001
1111
```

**Output**

```
0
```

**Input**

```
1
6 6 5 4
010101
111000
001010
111111
000000
101010
```

**Output**

```
5
```

ACCEPTED SOLUTION (C++)

```cpp
#include <bits/stdc++.h>
using namespace std;

// 2D prefix sum over a matrix of ints (0/1).
struct Pref2D {
    int n, m;
    vector<vector<int>> ps; // (n+1) x (m+1)
    Pref2D() {}
    Pref2D(int n_, int m_) : n(n_), m(m_), ps(n_+1, vector<int>(m_+1, 0))
        {}
    void build(const vector<vector<int>>& a) {
        for (int i = 0; i < n; ++i) {
            int rowSum = 0;
            for (int j = 0; j < m; ++j) {
                rowSum += a[i][j];
                ps[i+1][j+1] = ps[i][j+1] + rowSum;
            }
        }
    }
    int sum(int r1, int c1, int r2, int c2) const {
        if (r1 > r2 || c1 > c2) return 0;
        return ps[r2+1][c2+1] - ps[r1][c2+1] - ps[r2+1][c1] + ps[r1][c1];
    }
};

int main() {
    ios::sync_with_stdio(false);
    cin.tie(nullptr);
    int t;
```

```
29    if (!(cin >> t)) return 0;
30    while (t--) {
31        int n, m, a, b;
32        cin >> n >> m >> a >> b;
33        vector<string> s(n);
34        for (int i = 0; i < n; ++i) cin >> s[i];
35
36        vector<vector<int>> A(n, vector<int>(m, 0));
37        for (int i = 0; i < n; ++i)
38            for (int j = 0; j < m; ++j)
39                A[i][j] = (s[i][j] == '1');
40
41        vector<vector<int>> rowPS(n, vector<int>(m+1, 0));
42        vector<vector<int>> colPS(n+1, vector<int>(m, 0));
43        for (int i = 0; i < n; ++i) {
44            for (int j = 0; j < m; ++j) {
45                rowPS[i][j+1] = rowPS[i][j] + A[i][j];
46                colPS[i+1][j] = colPS[i][j] + A[i][j];
47            }
48        }
49
50        int NQn = max(0, n-1), NQm = max(0, m-1);
51        vector<vector<int>> Q(NQn, vector<int>(NQm, 0));
52        for (int i = 0; i+1 < n; ++i) {
53            for (int j = 0; j+1 < m; ++j) {
54                Q[i][j] = A[i][j] ^ A[i+1][j] ^ A[i][j+1] ^ A[i+1][j+1];
55            }
56        }
57        Pref2D Qps(NQn, NQm);
58        Qps.build(Q);
59
60        long long best = (long long)1e18;
61        int h = a, w = b;
62
63        for (int r = 0; r + h <= n; ++r) {
64            for (int c = 0; c + w <= m; ++c) {
65                int rq1 = r, cq1 = c, rq2 = r + h - 2, cq2 = c + w - 2;
66                int total = Qps.sum(rq1, cq1, rq2, cq2);
67                if (h >= 2 && w >= 2) {
68                    total -= Q[rq1][cq1];
69                    total -= Q[rq1][cq2];
70                    total -= Q[rq2][cq1];
71                    total -= Q[rq2][cq2];
72                }
73                if (total != 0) continue;
74
75                int pivotRow = r + 1, pivotCol = c + 1;
76                int p = A[pivotRow][pivotCol];
77
78                int innerColsOnes = rowPS[pivotRow][c + w - 1] - rowPS[
                      pivotRow][c + 1];
79                int leftEnd = A[pivotRow][c] ^ 1;
80                int rightEnd = A[pivotRow][c + w - 1] ^ 1;
81                int S_cols = innerColsOnes + leftEnd + rightEnd;
82
83                int L = r + 1, U = r + h - 2;
84                int len = max(0, U - L + 1);
85                int onesInSeg = (len ? (colPS[U+1][pivotCol] - colPS[L][
                      pivotCol]) : 0);
86                int unequalInterior = (p == 0 ? onesInSeg : (len -
                      onesInSeg));
87
88                int d_top = A[r][pivotCol] ^ p;
89                int d_bot = A[r + h - 1][pivotCol] ^ p;
90                int S_rows = unequalInterior + 2 - d_top - d_bot;
```

```
91
92                  int S0 = S_rows + S_cols;
93                  int flips = min(S0, h + w - S0);
94                  best = min(best, (long long)flips);
95              }
96          }
97
98          if (best == (long long)1e18) cout << -1 << "\n";
99          else cout << best << "\n";
100     }
101     return 0;
102 }
```

## C  EXAMPLE OF LLM GENERATED PROBLEM THAT ITSELF CANNOT SOLVE: IT RESTAURANTS — DISTANCE-2 RIVALRY

**Time limit per test:** 2 seconds
**Memory limit per test:** 256 megabytes

City $N$ still has a tree road network with $n$ junctions ($n-1$ roads, connected, undirected). The mayor again wants to place restaurants of two rival IT networks: iMac Donalds and Burger Bing. Each junction can host at most one restaurant.

This time, the owners impose a stricter rule set:

- Rivalry (distance-1): Two neighboring junctions (joined by a road) cannot host restaurants of different networks.

- Brand dilution (distance-2): Two junctions at distance exactly 2 (i.e., they have a common neighbor) cannot host restaurants of the same network.

Both networks must build at least one restaurant. The mayor wants to maximize the total number of restaurants. Among all optimal placements, find all pairs $(a, b)$ such that $a$ restaurants belong to Mac Donalds and $b$ to Burger Bing, with $a + b$ maximized.

Print all such pairs sorted by increasing $a$.

### INPUT

The first line contains a single integer $n$ ($3 \leq n \leq 2000$).

Then follows $n-1$ lines: edges $x_i\ y_i$ ($1 \leq x_i,\ y_i \leq n$).

The edges form a tree.

### OUTPUT

```
z
a1 b1
a2 b2
...
az bz
```

Where $z$ is the count of pairs, and $(a_i, b_i)$ are all pairs with maximum $a_i + b_i$, sorted by ai ascending. If no valid placement uses both networks, print 0 only.

### EXPLANATION OF THE NEW RULES

- Adjacent restaurants cannot be different brands (so if both ends of an edge have restaurants, they must be the same brand).

- However, for any chain $u - v - w$, if $u$ and $w$ both have restaurants, then they must be different brands (distance-2 same-brand is forbidden).
- Together these imply strong local coupling between siblings in the rooted tree:
  - If a node is occupied, then at most one of its children can also be occupied at its root (and it must have the same brand as the parent).
  - If a node is empty, then among its children, at most one may have a red root and at most one may have a blue root (siblings with occupied roots must have different brands).

These constraints make the original "remove one vertex and do subset sums" approach invalid; we need a new DP.

EXAMPLES

**Input**

```
5
1 2
2 3
3 4
4 5
```

**Output**

```
2
1 2
2 1
```

One optimal solution uses 3 restaurants on the path (e.g., 1=R, 2=R, 4=., 5=B), giving totals $(2, 1)$ or $(1, 2)$.

**Input**

```
5
1 2
1 3
1 4
1 5
```

**Output**

```
1
1 1
```

A star: the center must be empty to allow both brands on leaves; at most one red leaf and one blue leaf $\rightarrow$ exactly $(1, 1)$.

## D   NEAR-VALID TEST CASE EXAMPLE: PERPENDICULAR SEGMENTS time

**time limit per test:** 2 seconds
**memory limit per test:** 256 megabytes

You are given a coordinate plane and three integers $X, Y$, and $K$. Find two line segments $AB$ and $CD$ such that

1. the coordinates of points $A, B, C$, and $D$ are integers;
2. $0 \leq A_x, B_x, C_x, D_x \leq X$ and $0 \leq A_y, B_y, C_y, D_y \leq Y$;
3. the length of segment $AB$ is at least $K$;
4. the length of segment $CD$ is at least $K$;

5. segments $AB$ and $CD$ are perpendicular: if you draw lines that contain $AB$ and $CD$, they will cross at a right angle.

Note that it's **not** necessary for segments to intersect. Segments are perpendicular as long as the lines they induce are perpendicular.

INPUT

The first line contains a single integer $t$ ($1 \leq t \leq 5000$) — the number of test cases. Next, $t$ cases follow.

The first and only line of each test case contains three integers $X, Y$, and $K$ ($1 \leq X, Y \leq 1000$; $1 \leq K \leq 1414$).

*Additional constraint on the input:* the values of $X$, $Y$, and $K$ are chosen in such a way that the answer exists.

OUTPUT

For each test case, print two lines. The first line should contain 4 integers $A_x, A_y, B_x$, and $B_y$ — the coordinates of the first segment.

The second line should also contain 4 integers $C_x, C_y, D_x$, and $D_y$ — the coordinates of the second segment.

If there are multiple answers, print any of them.

EXAMPLE

**input**

```
4
1 1 1
3 4 1
4 3 3
3 4 4
```

**output**

```
0 0 1 0
0 0 0 1
2 4 2 2
0 1 1 1
0 0 1 3
1 2 4 1
0 1 3 4
0 3 3 0
```

D.1   NEAR-VALID TEST CASE EXAMPLE

Here are two minimal test inputs that differ by only 1 in a single field ($K$). The first is eligible (a solution exists), the second is not (no segment of length $\geq K$ can fit at all).

**Valid**

```
1
4 4 5
```

Why: In a 4×4 box, you can take two perpendicular diagonals: $(0,0) - (4,4)$ and $(0,4) - (4,0)$. Each has length $\sqrt{4^2 + 4^2} = \sqrt{32} \approx 5.657 \geq 5$.

**Near Valid**

```
1
4 4 6
```

Why: The longest possible segment inside a 4×4 box is the diagonal 5.657, which is $< 6$, so even one segment of length $\geq 6$ can't exist, hence two perpendicular ones can't either.

## E   EXAMPLE OF INCORRECT MUTANT PROGRAMS: GUESS THE TREE

**time limit per test:** 2 seconds
**memory limit per test:** 256 megabytes

*This is an interactive problem.*

Misuki has chosen a secret tree with $n$ nodes, indexed from 1 to $n$, and asked you to guess it by using queries of the following type:

- "? a b" — Misuki will tell you which node $x$ minimizes $|d(a, x) - d(b, x)|$, where $d(x, y)$ is the distance between nodes $x$ and $y$. If more than one such node exists, Misuki will tell you the one which minimizes $d(a, x)$.

Find out the structure of Misuki's secret tree using at most $15n$ queries!

### INPUT

Each test consists of multiple test cases. The first line contains a single integer $t$ ($1 \leq t \leq 200$) — the number of test cases.

Each test case consists of a single line with an integer $n$ ($2 \leq n \leq 1000$), the number of nodes in the tree.

It is guaranteed that the sum of $n$ across all test cases does not exceed 1000.

### INTERACTION

The interaction begins by reading the integer $n$.

Then you can make up to $15n$ queries.

To make a query, output a line in the format "? a b" (without quotes) ($1 \leq a, b \leq n$). After each query, read an integer — the answer to your query.

To report the answer, output a line in the format "! a1 b1 a2 b2 ... a_{n-1} b_{n-1}" (without quotes), meaning that there is an edge between nodes $a_i$ and $b_i$, for each $1 \leq i \leq n - 1$. You can print the edges in any order.

After $15n$ queries have been made, the response to any other query will be $-1$. Once you receive such a response, terminate the program to receive the Wrong_Answer verdict.

After printing each line, do not forget to output the end of line and flush the output buffer. Otherwise, you will receive the Idleness limit exceeded verdict. To flush, use:

- fflush(stdout) or cout.flush() in C++;

- System.out.flush() in Java;

- flush(output) in Pascal;

- stdout.flush() in Python;

- see the documentation for other languages.

HACKS

For hacks, use the following format: The first line contains an integer $t$ ($1 \leq t \leq 200$) — the number of test cases.

The first line of each test contains an integer $n$ — the number of nodes in the hidden tree.

Then $n - 1$ lines follow. The $i$-th of them contains two integers $a_i$ and $b_i$ ($1 \leq a_i, b_i \leq n$), meaning that there is an edge between $a_i$ and $b_i$ in the hidden tree.

The sum of $n$ over all test cases must not exceed 1000.

EXAMPLE

**input**

```
1
4
1
1
3
```

**output**

```
? 1 2
1
? 1 3
1
? 1 4
3
! 1 2 1 3 3 4
```

NOTE

A tree is an undirected acyclic connected graph. A tree with $n$ nodes will always have $n - 1$ edges.

In the example case, the answer to `"? 1 2"` is 1. This means that there is an edge between nodes 1 and 2.

The answer to `"? 1 3"` is 1. This means that there is an edge between nodes 1 and 3.

The answer to `"? 1 4"` is 3. It can be proven that this can only happen if node 3 is connected to both node 1 and 4.

The edges of the tree are hence $(1, 2)$, $(1, 3)$ and $(3, 4)$.

E.1  WHY IT'S WRONG

Here's a subtly broken version. It behaves the same on the sample, but the component update decision is based on comparing the node index to the label of the found node (instead of comparing component labels). This will make it fail on plenty of hidden cases while still looking plausible.

```
1  #include <iostream>
2  #include <numeric>
3  #include <vector>
4  #include <array>
5
6  using namespace std;
7
8  int query(int u, int v) {
9      cout << "? " << u + 1 << ' ' << v + 1 << '\n';
10     int x; cin >> x;
11     return x - 1;
```

```
12  }
13
14  int main() {
15    int t; cin >> t;
16    while (t--) {
17      int n; cin >> n;
18
19      vector<array<int, 2>> e;
20      vector<int> c(n);
21      iota(c.begin(), c.end(), 0);
22
23      auto addEdge = [&](int u, int v) {
24        e.push_back({u, v});
25        vector<int> cand;
26        for (int i = 0; i < n; i++)
27          if (c[i] == c[v])
28            cand.emplace_back(i);
29        for (int i : cand)
30          c[i] = c[u];
31      };
32
33      for (int i = 0; i < n - 1; i++) {
34        int u = 0, v = 0;
35        while (c[v] == c[u]) v++;
36        int x;
37        // SUBTLE BUG: use (u == c[x]) instead of (c[u] == c[x]).
38        while ((x = query(u, v)) != u) {
39          if (u == c[x]) u = x;   // <-- wrong criterion
40          else v = x;
41        }
42        addEdge(u, v);
43      }
44
45      cout << "!";
46      for (auto [u, v] : e) cout << ' ' << u + 1 << ' ' << v + 1;
47      cout << '\n';
48    }
49  }
```

# F   HUMAN EXPERT GRADING POLICY

Each problem is assessed independently across five dimensions: Solvability, Solution Correctness, Quality Rating, Novelty, and Difficulty.

## F.1   SOLVABILITY

This criterion evaluates whether a deterministic solution exists that satisfies the problem's time and space constraints.

- **Yes:** A deterministic solution exists that satisfies the problem's time and space constraints.
- **No:** It can be proven that no such solution exists.
- **Unknown:** It is unclear whether a valid deterministic solution exists.

## F.2   SOLUTION CORRECTNESS

This evaluates whether the provided reference solution (e.g., a model-generated code) is fundamentally correct in its approach.

- **Yes:** The core idea of the solution is correct. We allow minor implementation errors, as long as the overall idea is correct.

- **No:** The solution contains fundamental flaws in reasoning or design.

- **Unknown:** It is not possible to determine the correctness of the provided solution approach.

### F.3  QUALITY RATING

The overall quality of a problem is rated on a scale from 1 to 5, considering clarity, originality of approach, and implementation difficulty.

- **1:** Severely flawed. For example, unsolvable problems or incomprehensible descriptions.

- **2:** Low quality. For example, unclear statements, trivial or uninteresting solutions, or unverifiable correctness of the reference solution.

- **3:** Moderate quality. The problem is clearly solvable, but may be derivative, overly complicated, inelegant, or rely on obscure knowledge.

- **4:** High quality. The problem is clear, solvable, moderately difficult, and contains at least one insightful idea, though its overall structure might still feel somewhat formulaic or standard.

- **5:** Excellent quality. The problem is elegant, novel, and appropriately challenging. It is free from contrived traps and its solution is non-trivial, requiring genuine insights.

### F.4  NOVELTY

This measures whether the problem introduces a genuinely new challenge.

- **Yes:** The problem is novel and not seen in prior contests or problem archives, even for experienced setters.

- **No:** The problem duplicates or closely resembles existing ones.

- **Unknown:** It is unclear whether the problem is novel.

### F.5  DIFFICULTY

Difficulty is assessed using the Codeforces rating scale (e.g., 800, 1200, 1600, . . . ). This score is provided as a reference and does not directly influence the overall rating.

### F.6  OVERALL PROBLEM RATING

A composite grading system is defined as follows. A problem must satisfy the requirements of its assigned level and all lower levels.

- $\geq$ **0:** All evaluated problems.

- $\geq$ **1:** Solvability = Yes.

- $\geq$ **2:** Solution Correctness = Yes.

- $\geq$ **3:** Quality Rating $\geq$ 3.

- $\geq$ **4:** Quality Rating $\geq$ 4.

- $\geq$ **5:** Novelty = Yes.

- $\geq$ **6:** Quality Rating $\geq$ 5.

## G  SIMULATOR IN INTERACTOR

---

**Algorithm 5:** SIMULATE (in Interactor)

**Input:** Interactor $I$, solver $\mathcal{A}$, Validator $V^{\star}$
**Output:** Accepted/Rejected

1 Initialize protocol state; set time/memory limits; seed RNG deterministically
2 **while** session active **do**
3   $msg_I \leftarrow I(\text{state})$; **if** $\neg$ **WELLFORMED**$(msg_I)$:
4    **return** Rejected
5   $msg_A \leftarrow \mathcal{A}(msg_I)$; **if** $\neg$ **WELLFORMED**$(msg_A)$:
6    **return** Rejected
7   **if** **PROVIDESFINALANSWER**$(msg_A)$:
8    **if** $V^{\star}(\textbf{implied input}) \neq \textbf{valid}$:
9     **return** Rejected
10    **return** JUDGEFINAL$(I, msg_A)$
11   UPDATESTATE$(\text{state}, msg_I, msg_A)$
12 **return** Rejected

---

## H  LIMITATIONS AND FUTURE WORK

While AutoCode demonstrates a robust pipeline for generating and verifying competitive programming problems, we acknowledge several limitations that open up promising avenues for future research.

**The quality judgment gap.** A primary limitation is that LLMs currently lack a robust, human-aligned sense of problem quality. As shown in Finding 4 of Section 5, there is little correlation between LLM self-evaluations and the scores provided by human experts. This judgment gap means our system must still rely on human experts to identify truly high-quality, novel problems. This reliance on manual annotation is a bottleneck to a scalable, high-quality problem generation pipeline. A key future direction is to develop a specialized judge LLM, fine-tuned on expert-rated problem datasets to better align its assessments with human preferences, potentially serving as a reliable automated filter.

**The bottleneck for automated self-improvement.** AutoCode is ideally suited for creating a self-improvement loop, where an LLM could enhance its reasoning abilities by training on the problems it generates. However, deploying and testing such a system faces immediate practical barriers. The most capable problem-generating models like GPT-5 are closed-source and do not expose the APIs necessary for reinforcement learning. Even for powerful open-source models, scalable and stable RL frameworks are not yet mature to support fine-tuning at this scale to the best of our knowledge. A natural future direction is to focus on developing RL or self-play frameworks compatible with frontier LLMs to unlock the full potential of AutoCode.

**Advancing beyond recombinational novelty.** Our analysis indicates that LLMs currently tend to create new problems by combining existing algorithms rather than creating conceptually new problems that require subtle insights. Future research could explore novel prompting strategies or problem-generation pipelines to encourage greater conceptual leaps. Additionally, for valid problems that LLMs can generate but cannot solve themselves (i.e., their reference solution std.cpp is incorrect), AutoCode's strict verification protocol currently filters them out. A prominent future direction is to develop scalable methods for identifying these valid but unsolved questions without relying on manual check by human experts, which is an extremely labor-intensive process.

## I PROMPT TEMPLATE

---

**Checker Prompt Template**

This is a competitive programming problem statement and its accepted solution:

```markdown
{{problem statement}}
```
```cpp
{{accepted solution}}
```

Implement a testlib checker for this problem.

If a standard checker is sufficient, your response should contain only the name of the appropriate `.cpp` file from the list below.

- `fcmp.cpp`: Compares files line by line, doesn't ignore trailing whitespaces on each line.
- `lcmp.cpp`: Compares files line by line, ignoring trailing whitespace on each line.
- `hcmp.cpp`: Compares a single huge integer.
- `wcmp.cpp`: Compares sequences of tokens, ignoring trailing whitespace.
- `ncmp.cpp`: Compares sequences of integers, ignoring trailing whitespace.
- `rcmp4.cpp`: Compares sequences of real numbers with a precision of $10^{-4}$.
- `rcmp6.cpp`: Compares sequences of real numbers with a precision of $10^{-6}$.
- `rcmp9.cpp`: Compares sequences of real numbers with a precision of $10^{-9}$.
- `yesno.cpp`: Compares a single "yes" or "no" token, case-insensitively.

For example, reply "`wcmp.cpp`", do not include quotation marks and other content.

If the problem requires custom validation logic that a checker comparator cannot handle, your response should contain the checker enclosed in a single markdown block:

```cpp
//checker code here
```

Please follow these additional constraints:

- Always include `"testlib.h"` first, then `<bits/stdc++.h>`, and add `using namespace std`.
- All `testlib.h` functions are in the global namespace. Never use the `testlib::` prefix or `using namespace testlib;`.
- Initialize your checker with `registerTestlibCmd(argc, argv);`.
- Use the built-in `inf` (input), `ouf` (contestant output), and `ans` (jury answer) stream. Call methods on the correct stream, e.g., `long long n = inf.readLong();`, `string s = ouf.readToken();`. Do not create extra stream objects.
- When passing a stream to a helper function, declare the parameter as `InStream&`; e.g., `int readAns(InStream& stream)`.
- The optional message parameter of `stream.readLong()`, `stream.readToken()`, etc. must be a fixed literal string—printf-style placeholders like `%d` are not supported. Write `inf.readLong(0LL, n, "x");` and never `inf.readLong(0LL, n, "test case #%d", tc);`.
- Use `stream.readLong()` for 64-bit integers. Ensure all arguments are `long long` by using the LL suffix (e.g., `inf.readLong(1LL, 100LL);`). Do not use arguments of type `int` or `double`, such as `ans.readLong(-1e9, 1e9);`.

---

- Use `stream.readToken()` to read a string; do not use `readString()`. Also, do not add extra arguments inside `readToken(...)`, for example, do not write `inf.readToken("string")`.

- Do not use `ensure` or `ensuref`; instead, call `quitf(_ok, "...")` for correct answers and `quitf(_wa, "...")` for wrong answers. Do not use string concatenation to form the message; it must be a single string literal.

- To read a character from a specific set: first read it using `char c = stream.readChar();`, then validate it with `quitf`.

- Use `stream.readSpace()` for a single space and `stream.readEoln()` for an end-of-line. These separators are read automatically, so you only need to call them if you are about to call `stream.readChar()` immediately afterward.

- Do not use the ternary operator (`?   :`) for void functions like `stream.readEoln()`. Use a standard if/else block instead.

- The judging machine does not have boost-multiprecision installed; avoid including or relying on it.

- Never use non-existent functions like `stream.skipSpaces()`, `stream.peekChar()`, `stringToLongLong()`, or `getTestCases()`.

- Do not check whether the stream has reached EOF; testlib handles end-of-file checks automatically. For example, avoid calling `ouf.readEof()`, `ans.readEof()`, `ouf.ensureEof()`, `ans.ensureEof()`, `ouf.afterEOF()`, or `ans.afterEOF()`.

- The `ans` stream is the output of the accepted solution. Do not skip any numbers or strings, and do not expect any extra content.

## Checker Tests Template

This is a competitive programming problem statement and its accepted solution:

````
```markdown
{{problem statement}}
```
```cpp
{{accepted solution}}
```
````

Write 20 checker tests for this problem.

Your response should contain the checker tests enclosed in a single markdown code block.
Use `"verdict":   1` when the output meets the problem constraints, and `"verdict":   0` otherwise.
Provide 5 correct outputs and 15 incorrect outputs.

`input` is valid input, `output` is the contestant's output, `answer` is the jury's output, and `verdict` indicates whether the output meets the problem's constraints.

The format should look like this:

```json
[
  { "input": "...", "output": "...", "answer": "...", "verdict": 1 },
  { "input": "...", "output": "...", "answer": "...", "verdict": 0 },
  ...
]
```

Please follow these additional constraints:

- Do not include any explanations, comments, or any other text outside of the JSON structure.
- The incorrect output should be an input that becomes correct with minor modifications.
- The answer you generate should preferably be the accepted solution that provides the correct output for the input.

## Generator Template 1

This is a competitive programming problem statement and its accepted solution:

```markdown
{{problem statement}}
```
```cpp
{{accepted solution}}
```

Implement a testlib generator for this problem. Your response should contain the generator enclosed in a single markdown block:

```cpp
//generator code here
```

The generator should take a single command-line argument (from 1 to 10) to determine which test case to generate.

**For a single-test problem:**
The generator must output exactly 10 small, distinct, and meaningful corner cases.

**For a multi-test problem (with $t$ test cases):**
Let $t_{\max}$ be the maximum allowed number of test cases. The generator must define 10 distinct test groups, each associated with a different command-line argument (from 1 to 10), and each with its own specific variable constraints (e.g., Group 1: $1 \le n \le 5, 1 \le a[i] \le 9$; Group 2: $1 \le n \le 10, 1 \le a[i] \le 3$).
Each group must be able to generate more than $t_{\max}/5$ but no more than $t_{\max}$ valid and distinct test cases.
The generator must then output exactly $t_{\max}$ test cases, randomly sampled from the distribution defined by the specified group.

Please follow these additional constraints:

- Always include `"testlib.h"` first, then `<bits/stdc++.h>`, and add `using namespace std`.
- All `testlib.h` functions are in the global namespace. Never use the `testlib::` prefix or `using namespace testlib;`.
- Never use the `global::` prefix or `std::`.
- Please use `int testId = atoi(argv[1]);` to obtain the test ID.
- Initialize your generator with `registerGen(argc, argv, 1);`.
- For conditional validation with a custom message, use `ensuref(condition, "message");`.
- When calling `rnd.next(l, r)`, ensure that both `l` and `r` are of the same type, either both `int` or both `long long`. Note that `rnd.next<int>(l, r)` is not a valid syntax.
- Use the global `shuffle(vec.begin(), vec.end());`. Do not use `rnd.shuffle(vec.begin(), vec.end())` or `shuffle(vec.begin(), vec.end(), rnd)`.

- Don't use `int64` or `i64`, that's not valid syntax.
- Don't define `lowerCase`, it will conflict with `std`. Use a different function name instead.
- Use `cout` to print all output.
- Do not set the random seed manually.
- Do not use non-existent methods like `rnd.sample()`, `rnd.permutation()`, `rnd.nextf()`, `randomString()`, or `global_shuffle()`.

## Generator Template 2

This is a competitive programming problem statement and its accepted solution:

```markdown
{{problem statement}}
```
```cpp
{{accepted solution}}
```

Implement a testlib generator for this problem. Your response should contain the generator enclosed in a single markdown block:

```cpp
//generator code here
```

The generator should take a single command-line argument (from 1 to 10) to determine which test case to generate. The generation strategy is divided into two parts:

**Large Random Cases (5 test cases)**
The goal is to fail incorrect solutions with strong, randomized data.

- Tests 1–2: Generate cases where $n$ and $t$ are large and roughly equal, with their product $n \times t$ approaching the maximum allowed limit.
- Tests 3–5: Generate cases where $t = 1$ and $n$ is set to its maximum possible value.

For all 5 tests:

- Some cases should be completely random, while others should conform to a specific structure relevant to the problem (e.g., a star graph, a chain, an array with all equal elements).
- While meeting the primary condition for the test (e.g., $n = \max$), randomize all other variables across their largest possible valid ranges.
- Ensure these cases collectively cover various scenarios that a correct solution must handle.

**Boundary/Edge Cases (5 test cases)**
The goal is to target specific, extreme, or tricky corner cases.
Analyze the given problem and select the 5 most effective strategies from the list below to create test cases that are most likely to fail incorrect solutions. You may also devise your own problem-specific strategies if they are more effective.

*Potential Strategies:*

- A test case that maximizes the algorithm's runtime.
- A test case that maximizes the numerical output value to check for integer overflows (e.g., `int`, `long long`).
- A test case that maximizes memory usage.
- A test case designed to cause hash collisions for common hashing methods.

- A test case that maximizes the total number of characters in the input or output.
- Problem-specific corner cases.

Please follow these additional constraints:

- Always include `"testlib.h"` first, then `<bits/stdc++.h>`, and add `using namespace std`.
- All `testlib.h` functions are in the global namespace. Never use the `testlib::` prefix or `using namespace testlib;`.
- Never use the `global::` prefix or `std::`.
- Please use `int testId = atoi(argv[1]);` to obtain the test ID.
- Initialize your generator with `registerGen(argc, argv, 1);`.
- For conditional validation with a custom message, use `ensuref(condition, "message");`.
- When calling `rnd.next(l, r)`, ensure that both `l` and `r` are of the same type, either both `int` or both `long long`. Note that `rnd.next<int>(l, r)` is not a valid syntax.
- Use the global `shuffle(vec.begin(), vec.end());`. Do not use `rnd.shuffle(vec.begin(), vec.end())` or `shuffle(vec.begin(), vec.end(), rnd)`.
- Don't use `int64` or `i64`, that's not valid syntax.
- Don't define `lowerCase`, it will conflict with `std`. Use a different function name instead.
- Use `cout` to print all output.
- Do not set the random seed manually.
- Do not use non-existent methods like `rnd.sample()`, `rnd.permutation()`, `rnd.nextf()`, `randomString()`, or `global_shuffle()`.

## Generator Template 3

This is a competitive programming problem statement and its accepted solution:

```markdown
{{problem statement}}
```
```cpp
{{accepted solution}}
```

Implement a testlib generator for this problem. Your response should contain the generator enclosed in a single markdown block:

```cpp
//generator code here
```

The generator should take a single command-line argument (from 1 to 10) to determine which test case to generate, to select which of the following ten test case generation strategies to use. You may also devise your own problem-specific strategies if they are more effective.

1. Generate a test with the maximum number of test cases ($t$), where $t - 1$ cases have minimal input size (e.g., $n = 1$), and one case has the largest possible $n$ to maximize the total input size.
2. Generate a test with the maximum number of test cases ($t$), where $n$ is the same large value for all cases, maximizing the total input size.

3. Generate a test where the output for each case is maximized, leading to a large total output size.

4. Generate a test case containing a long, contiguous sequence of identical values.

5. Generate a test case containing a long, strictly increasing sequence.

6. Generate a test case containing a long, strictly decreasing sequence.

7. Generate a test case containing a long sequence of alternating large and small values.

8. Generate a test case designed to exploit the worst-case time complexity of a common brute-force or incorrect algorithm for this problem.

9. Generate a problem-specific test case that maximizes a particular metric or constraint defined in the problem statement, pushing it to its limits.

10. Generate a test case that is unlikely to be covered by small or random inputs, targeting specific edge conditions that might lead to performance issues.

Please follow these additional constraints:

- Always include `"testlib.h"` first, then `<bits/stdc++.h>`, and add `using namespace std`.

- All `testlib.h` functions are in the global namespace. Never use the `testlib::` prefix or `using namespace testlib;`.

- Never use the `global::` prefix or `std::`.

- Please use `int testId = atoi(argv[1]);` to obtain the test ID.

- Initialize your generator with `registerGen(argc, argv, 1);`.

- For conditional validation with a custom message, use `ensuref(condition, "message");`.

- When calling `rnd.next(l, r)`, ensure that both `l` and `r` are of the same type, either both `int` or both `long long`. Note that `rnd.next<int>(l, r)` is not a valid syntax.

- Use the global `shuffle(vec.begin(), vec.end());`. Do not use `rnd.shuffle(vec.begin(), vec.end())` or `shuffle(vec.begin(), vec.end(), rnd)`.

- Don't use `int64` or `i64`, that's not valid syntax.

- Don't define `lowerCase`, it will conflict with `std`. Use a different function name instead.

- Use `cout` to print all output.

- Do not set the random seed manually.

- Do not use non-existent methods like `rnd.sample()`, `rnd.permutation()`, `rnd.nextf()`, `randomString()`, or `global_shuffle()`.

---

## Validator Template

This is a competitive programming problem statement and its accepted solution:

```markdown
{{problem statement}}
```
```cpp
{{accepted solution}}
```

Implement a testlib validator for this problem. Your response should contain the validator enclosed in a single markdown block:

```cpp
```

```
//validator code here
```

Please follow these additional constraints:

- Always include "testlib.h" first, then <bits/stdc++.h>, and add using namespace std.

- All testlib.h functions are in the global namespace. Never use the testlib:: prefix or using namespace testlib;.

- Initialize your validator with registerValidation(argc, argv);.

- All validator testlib read functions are methods of the inf object. Always call them using inf., for example: inf.readInt(). Do not use ouf or ans.

- Use inf.readLong() for 64-bit integers. Ensure all arguments are long long by using the LL suffix (e.g., inf.readLong(1LL, 100LL);).

- Use inf.readToken() to read a string; do not use readString(). Also, do not add extra arguments inside readToken(...), for example, do not write inf.readToken("string").

- To read a character from a specific set: first read it using char c = inf.readChar();, then validate it with ensuref, e.g., ensuref(c == '+' || c == '-', "Must be + or -");.

- For conditional validation with a custom message, use ensuref(condition, "message");.

- Always read separators explicitly. Use inf.readSpace() for a single space, inf.readEoln() for end-of-line, and inf.readEof() for end-of-file. These calls are mandatory.

- The last two reads are always inf.readEoln() and then inf.readEof().

- Do not use the ternary operator (?   :) for void functions like inf.readEoln(). Use a standard if/else block instead.

- The judging machine does not have boost-multiprecision installed; avoid including or relying on it.

- Never use non-existent functions like inf.skipSpaces() or inf.peekChar().

- Do not use quit or quitf.

- If the input violates the additional constraints guaranteed by the problem statement (e.g., an answer is guaranteed to exist), it must be judged as invalid.

- If the input would cause the accepted solution to fail, the validator must reject it.

## Validator Tests Template

This is a competitive programming problem statement and its accepted solution:

```markdown
{{problem statement}}
```
```cpp
{{accepted solution}}
```

Write 20 validator tests for this problem.

Your response should contain the validator tests enclosed in a single markdown code block. Use "verdict": 1 when the input meets the problem constraints, and "verdict": 0 otherwise.
Provide 5 valid inputs and 15 invalid inputs.

The format should look like this:

```json
[
  { "input": "...", "verdict": 1 },
  { "input": "...", "verdict": 0 },
  ...
]
```

Please follow these additional constraints:

- Do not include any explanations, comments, or any other text outside of the JSON structure.
- The invalid input should be an input that becomes valid with minor modifications.
- If the problem includes additional constraints, such as "an answer is guaranteed to exist", prioritize constructing invalid cases that meet other conditions but not the additional constraint.
- If the input would cause the accepted solution to fail, it must be invalid.

---

### Author Template

This is a competitive programming problem statement and its accepted solution:

```markdown
{{problem statement}}
```
```cpp
{{accepted solution}}
```

You are an experienced competitive programming problem setter. Please modify at least one condition of the problem to design a new, more difficult problem. This new problem must require a different solution approach from the original and should not be solvable using heuristic methods. The problem must be of such quality and novelty that it could be accepted on a platform like Codeforces, or at the very least serve as a valuable training exercise. In the problem statement, please include time and space constraints and provide sample test cases. Furthermore, please write an accepted C++ solution, along with a complexity analysis, a proof of correctness, and a clear explanation.

**Response Format:**

```markdown
// Problem statement for the new problem
```
```cpp
// Accepted solution for the new problem
```
// Solution explanation and proof

---

## J  EARLY RESULTS OF RL ON AUTOCODE DATA

To provide an objective complement to our human evaluation of the "Model-Training Usable" criterion, we conduct a preliminary RL experiment using `Qwen3-8B-Thinking` as the base model. We employ the Slime framework[1] to implement GRPO, using the 133 AutoCode-generated problems as training data and the corresponding AutoCode-generated test cases as the reward signal. The model is trained for 5 epochs with a global batch size of 64, a constant learning rate of $1 \times 10^{-6}$,

---

[1] https://github.com/THUDM/slime

and the AdamW optimizer ($\beta_1 = 0.9$, $\beta_2 = 0.98$). During the rollout phase, we generate 8 samples per prompt with temperature $0.8$. We then evaluate the model on LiveCodeBench before and after fine-tuning.

| Model | Metric | Base Model | Fine-tuned w/ AutoCode |
|-------|--------|------------|------------------------|
| **Qwen3-8B** | Pass@1 | 46.2% | **48.5%** |

Table 3: Preliminary GRPO fine-tuning on AutoCode-generated problems and test cases. Training on AutoCode data yields a improvement in LiveCodeBench Pass@1, providing initial evidence that the generated problems are useful for model training.

This experiment is intentionally small-scale and serves as a proof-of-concept; scaling up the training (e.g., more problems, longer training, and additional base models) is left for future work to more rigorously quantify the utility of AutoCode-generated problems for model training.

## K  COMPARISON WITH HARDTESTS

To demonstrate that the performance gains are derived from our proposed AutoCode pipeline rather than simply the use of better models (e.g., o3, GPT-5), we conducted a controlled quantitative analysis. We compared AutoCode directly against HardTests, keeping all other variables constant. In particular, we use the exact same LLMs for generation (GPT-4o and o3) and identical execution environments for both pipelines. We use the official HardTests implementation (https://github.com/LeiLiLab/HardTestGen). The results (shown in the table below) indicate that AutoCode outperforms HardTests on both the GPT-4o model and the more advanced o3 model. We focus on comparing with HardTests for a few reasons: (i) CodeContests+ does not disclose the specific models used for generation, nor is their pipeline open-source, making a fair, controlled comparison impossible. (ii) The other two baselines we compare with, TACO and CodeContests assume the existence of and rely on a large number of correct submissions to stress test the legitimacy of test cases. This is an unrealistic assumption when creating test cases for new problems. Our pipeline operates without this dependency, which we believe is a stronger and more practical setting. (iii) HardTests has the strongest performance (highest consistency) out of the four baselines we compared against in the paper.

| Pipeline | Model | Checker Model | Consistency (Acc) ($\uparrow$) | FPR ($\downarrow$) | FNR ($\downarrow$) |
|----------|-------|---------------|-------------------------------|--------------------|--------------------|
| HardTests | 4o | AutoCode (4o) | 78.4% | 12.2% | **31.4%** |
| **AutoCode (Ours)** | 4o | AutoCode (4o) | **79.5%** | **7.5%** | 34.2% |
| HardTests | o3 | AutoCode (o3) | 80.0% | 21.5% | 18.5% |
| **AutoCode (Ours)** | o3 | AutoCode (o3) | **91.1%** | **3.7%** | **14.1%** |

Table 4: Fair comparison with Hardtests.

On the 4o model: AutoCode demonstrates higher consistency (79.5% vs 78.4%) and a lower False Positive Rate compared to the baseline. On the o3 model: The advantage of AutoCode is significant, with accuracy increasing to 91.1% (compared to 80.0% for the baseline) and the False Positive Rate (FPR) dropping to an extremely low 3.7%. Special Note on Experimental Setup: In the comparison experiments, the native Checkers generated by HardTests were not in the standard testlib.h format, meaning they could not be run directly on standard Online Judge systems. To ensure fairness and reproducibility, we used the standard-compliant Checkers generated by our AutoCode pipeline for all experiments listed above.

## L  HUMAN EVALUATION PROCESS

To assess the quality of the LLM-generated problems beyond automated metrics, we conducted a structured human evaluation with experienced competitive programmers.

**Expert background**   We invite 5 human experts to evaluate the LLM-generated problems. The group consisted of 2 Codeforces International Grandmasters (reviewing high-difficulty problems) and 3 Codeforces Masters (reviewing medium-to-low difficulty problems).

**Evaluation time**   Our experts comprehensively evaluate each problem's solvability, difficulty, novelty, quality, and the correctness of the LLM-provided solution. The average time spent reviewing each problem was approximately 30 minutes.

**Sample size**   We generate and review a total of 190 new problems, which is the data source for Figure 3. Within these, 133 problems were generated using our final prompt version, and these constitute the dataset for Figure 4. We will add these key numbers to the appendix of the paper to improve transparency and credibility.

