# OpenReview forum: "AutoCode: LLMs as Problem Setters for Competitive Programming"
_ICLR.cc/2026/Conference — ICLR 2026 Poster_

### Official Review · Reviewer_gTC1 · 2025-11-01

**Soundness:** 3
**Presentation:** 3
**Contribution:** 3
**Rating:** 6
**Confidence:** 3

**Summary:**

The paper focuses on test generation and problem creation for competitive programming problems. First, they show that their test generation framework can achieve very low false positive and false negative rates compared to prior SOTA work. The test generation framework consists of using LLMs to construct validator, generator, and checker with novel prompting techniques specifically designed for more robust checking. Building on the robust test generation, they show that they can prompt LLMs to create very high quality problems (judged by top competitive programmers) by seeding from existing problems and then going through testing to filter problems such that the LLM-generated brute-force solution (acting as ground truth) and the LLM-generated solution agree.

**Strengths:**

- Significantly improve the test generation false positive and false negative rate for the competition programming
- Demonstrate that model can generate novel problems not only useful for training but also can generate very high quality deemed as good as top competition problems (ICPC/IOI-Level)
- Analysis and ablation showing many insights of the results, the finding from the human experts annotation is especially valuable

**Weaknesses:**

The critical weakness of this paper is the lack of reproducibility. The paper uses specific LLM prompting techniques to achieve impressive results but fails to provide any of the prompts used in the work. Many details are also missing, for example, how the generator works and the problem creation procedure are not clearly explained.

**Questions:**

- Although stated in Finding 1: "LLMs can generate solvable problems that they themselves are unable to solve," this kind of problem would be filtered by the solvability check right? Could you clarify if the system using the filtering process described in the beginning of Section 5 would be able to produce such problems at the end?
- What is the "prompt optimization" refer to? Line 291 shows the ablation "without prompt optimization" and in line 321 "Prompt optimization turns out to be especially important;" there seems to be no explanation of the prompt optimization in the paper.
- In 3.2 for the generator, it is stated that they use strategies: Small Data Exhaustion, Randomized and Extreme Data, and TLE-Inducing Data, but didn't explain how these strategies are implemented. Are they being implemented with LLM by using LLM to generate a program to perform this kind of strategy?
- Can you explain GENERATECHECKERSCENARIOS in Section 3.3? How does it generate the (input, contestant_out, ref_out, verdict) data?
- Can you clarify which parts of the test generation required access to ground-truth programs? is it just the for thchecker?
- In line 285 "all results in this table are generated by GPT-5-High", do you mean the model used in the test generation system or the 33 submissions?
- Since competition programs accept or not sometimes depending on the timeout set for the program, how is timeout set for the generated problems?
- For the problem creation prompt that prompts the model to create new problems from seed problems, do you need to specifically guide the model what kind of variations they can make and does it bias the model to alter the seed problem in specific ways?
- Are the LLM prompting used in this work zero-shot or few-shot like you need to provide more examples to guide the model? Can you provide the prompts used in this work to help with reproducibility?

---

> ### Author Response · Authors · 2025-11-21
>
> We thank the reviewer for the positive assessment of our work's soundness and contribution. We fully agree that reproducibility is critical. Below, we address your specific questions regarding prompts, implementation details, and methodologies.
>
> ---
>
> **Q1. Reproducibility and Prompts**
>
> We completely agree that prompts are central to our work. To ensure full transparency, we have provided all prompt templates in our anonymous repository: https://anonymous.4open.science/r/sfbgtqwerxzcwes-5BB0. We commit to open-sourcing the full codebase upon acceptance.
> You can find specific files including:
> - test_generation/checker/checker_template.txt
> - test_generation/generator/generator_template[1-3].txt
> - test_generation/validator/validator_template.txt
> - test_generation/author/author_template.txt
>
> ---
>
> **Q2. Zero-shot vs. Few-shot**
>
> Our approach is essentially Zero-shot, but one enriched with highly detailed instructions and API constraints. We attempted traditional Few-shot methods but found them less effective; the model tended to overfit the examples (mimicking specific syntax rigidly) rather than using the appropriate testlib.h APIs flexibly for new contexts. Our detailed Zero-shot approach proved more robust for adhering to strict library specifications.
>
> ---
>
> **Q3. Finding 1 vs. Solvability Check**
>
> Your understanding is correct. If the LLM-generated std.cpp fails the cross-validation against brute.cpp, the problem is indeed filtered out by the Solvability Check described in Section 5.
> Finding 1 highlights a crucial gap: the LLM can conceive a logically valid and solvable problem (verified by humans) but often fails to implement the correct solution itself. Note that cases where the LLM generates a valid problem but an incorrect solution are valuable "gold data" for self-improvement training.
>
> ---
>
> **Q4. Explanation of "Prompt Optimization"**
>
> "Prompt Optimization" refers to our iterative process of analyzing failure cases to systematically tune the instructions. For instance, in checker_template.txt, we explicitly forbid ensure/rnd.shuffle() (requiring quitf/global shuffle instead) to satisfy testlib.h constraints. As shown in the Ablation Study (Table 2), removing these optimized constraints leads to a significant rise in False Negative Rate (from 1.2% to 2.9%) due to compilation errors and non-compliance.
>
> ---
>
> **Q5. Generator Strategies Implementation**
>
> Yes, these strategies are implemented by prompting the LLM to write specific C++ generator programs. You can see in generator_template[1-3].txt how we guide the LLM to write code for "Small Data," "Randomized/Extreme Data," and "TLE-Inducing Data" respectively. This approach yields higher correctness and better adversarial cases than direct data generation.
>
> ---
>
> **Q6. Logic of generate checker scenarios**
>
> This function creates a high-quality evaluation set ($Q$) to test candidate Checkers (see checker_tests_template.txt):
>
> **Task:** The LLM is given the problem and reference solution and asked to generate a list of scenarios.
>
> **Structure:** Each scenario is a tuple: (input, contestant_output, reference_output, expected_verdict).
>
> **Composition:** We execute the prompt twice to generate **40 scenarios** total. Crucially, this includes **30 negative cases** (verdict 0) where the LLM generates subtle errors (e.g., off-by-one, precision, formatting) and **10 positive cases** (verdict 1).
>
> **Selection:** The candidate Checker that best reproduces the expected_verdict for these 40 scenarios is selected.
>
> ---
>
> **Q7. Dependency on Ground Truth**
>
> This is a key design decision. We provide the ground-truth program to all components (Validator, Generator, Checker, Interactor) to maximize quality:
>
> **Validator:** Needs to understand the solution logic to filter out inputs that are mathematically impossible to solve.
>
> **Generator:** Needs the solution to construct adversarial (TLE-inducing) cases.
>
> **Checker:** Analyzing the reference logic simplifies the verification of complex outputs.
>
> **Interactor:** The reference solution is essential for mutation-based testing.
>
> ---
>
> **Q8. Clarification of Line 285 (GPT-5-High)**
>
> "GPT-5-High" refers to the model used to operate the AutoCode framework (i.e., generating the test components). The "33 submissions" serve as the evaluation dataset (simulating contestant submissions of varying quality). Line 285 indicates that the results in Table 2 reflect the performance of the AutoCode pipeline when driven by GPT-5-High.

---

> > ### Author Response · Authors · 2025-11-21
> >
> > **9. Setting Timeouts**
> >
> > Time limits are set dynamically based on the reference solution (std.cpp):
> > We measure the maximum execution time ($T_{max}$) of std.cpp across all generated tests.
> > We set the limit as $T_{final} = \lceil T_{max} \times 3 \rceil$.
> > The time is capped at a maximum of 10 seconds.
> >
> > ---
> >
> > **Q10. Problem Creation and Bias**
> >
> > In author_template.txt, we balance quality and innovation:
> >
> > **Guidance:** We explicitly require the new problem to be **strictly harder**, require a **different solution approach**, and be **unsolvable by heuristics**.
> >
> > **Avoiding Bias:** We do not dictate how to modify the problem (e.g., we do not say "change tree to graph"), only that they must "modify at least one condition." This "high standard, low restriction" approach fosters diversity among the new problems.

---

> > > ### Comment · Reviewer_gTC1 · 2025-11-27
> > >
> > > Thank you for taking time to address the comments. I have raised the score.

---

### Official Review · Reviewer_hPcp · 2025-11-02

**Soundness:** 2
**Presentation:** 4
**Contribution:** 2
**Rating:** 2
**Confidence:** 4

**Summary:**

This paper presents AutoCode, an LLM agent system for writing and developing competitive programming problems. AutoCode generally consists of two parts: a test case generator, and a problem writer.  The test generation component implements multiple components including validator, generator, checker, and interactor. Experimental results indicate that the test cases generated by AutoCode outperform current state-of-the-art baselines in terms of both FPR and FNR metrics. The problem writing component has achieved excellent results in human evaluation, and the findings show that LLMs are capable of generating high-quality and high-difficulty problems.

**Strengths:**

1. In terms of test case generation, AutoCode has indeed gone deeper and more refined than previous work (e.g., HardTests, CodeContests+). For instance, it enhances the validator by adding validator tests to further ensure the validator’s correctness, and it is capable of writing interactors for interactive problems.
2. This paper conducts pioneering research on problem writing using LLMs and validates the problem-writing capabilities of LLMs through large-scale human evaluations.
3. The study on the value alignment between LLMs and human experts is highly insightful. Given that LLM-as-a-judge is currently a highly mainstream method, the results (Fig. 4) demonstrate the limitations of LLMs in such complex OOD tasks.
4. This paper is well-written. I am enjoying reading this paper.

**Weaknesses:**

1. **Does this paper propose a better test case construction method?**, The generator-validator-checker-interactor workflow is a widely adopted problem-construction process in competitive programming, and has also been used more or less in previous studies (e.g., HardTests, CodeContests+). The similarity between AutoCode’s overall workflow and that of previous works may raise questions about the source of improvements in its evaluation. First, this study is conducted based on SOTA models such as o4 and GPT-5, while previous works are before the release of such advanced models. Second, prompts are a critical factor affecting performance—providing a open agent framework that includes prompts would be a significant contribution to the community. However, this paper does not disclose the prompts it used, thus diminishing its reproducibility and overall contribution.
2. **Questions on the soundness of the LLM problem writing study.** I find the topic of LLM problem writing very interesting, but I do not believe this paper has conducted an in-depth and rigorous study. First, it is an experiment entirely based on human evaluations, yet I think some of the criteria could have been tested through objective experiment. For instance, the paper mentions a criterion called "Model-Training Usable"—this criterion could be fully verified through actual training results, rather than relying on human evaluations. Additionally, the human evaluation lacks details, such as the average time each expert spent evaluating each piece of data and the profiles of the experts. Some conclusions also lack sufficient evidential support; for example, for Findings 2, it is recommended to incorporate more case studies for further discussion. Furthermore, the sample size of this experiment should be placed in a more prominent position, such as the caption of Figure 3. I read through this section several times but still could not find this number. Finally, regarding the most critical part of this experiment: how did the authors guide LLMs to generate problems? The entire problem-writing process lacks a detailed description,  nor does the prompt provided, so I believe it lacks reproducibility.

**Questions:**

1. How do you develop the sandbox system? Submissions in CodeContests may come from many different online judge systems, using a variety of operating systems (Linux, Windows), compilers (GCC and MSVC spanning multiple eras), and interpreters (Python 2, Python 3, and JVMs of various versions). Therefore, if you want to evaluate the consistency with official verdicts, the first thing to do is achieve environment alignment—and I think this is extremely difficult. Many submissions fail to produce the same verdicts even when using the exact same original test cases. How do you solve this problem?

---

> ### Author Response · Authors · 2025-11-21
>
> We sincerely thank the reviewer for the detailed feedback and for recognizing the strengths of our presentation. We value your questions regarding baselines, reproducibility, and evaluation details, and we address them point-by-point below.
>
> ---
>
> **Q1. Comparison with Baselines (HardTests)**
>
> To demonstrate that the performance gains are derived from our proposed AutoCode pipeline rather than simply the use of better models (e.g., o3, GPT-5), we conducted a controlled quantitative analysis. We compared AutoCode directly against HardTests (He et al., 2025), keeping all other variables constant. In particular, we use the exact same LLMs for generation (GPT-4o and o3) and identical execution environments for both pipelines. We use the official HardTests implementation (https://github.com/LeiLiLab/HardTestGen). The results (shown in the table below) indicate that AutoCode outperforms HardTests on both the GPT-4o model and the more advanced o3 model.
> We focus on comparing with HardTests for a few reasons: (i) CodeContests+ (Wang et al., 2025) does not disclose the specific models used for generation, nor is their pipeline open-source, making a fair, controlled comparison impossible. (ii) The other two baselines we compare with, TACO (Li et al., 2023) and CodeContests (Li et al., 2022) assume the existence of and rely on a large number of correct submissions to stress test the legitimacy of test cases. This is an unrealistic assumption when creating test cases for new problems. Our pipeline operates without this dependency, which we believe is a stronger and more practical setting. (iii) HardTests has the strongest performance (highest consistency) out of the four baselines we compared against in the paper.
>
> | Pipeline | Model | Checker Model | Consistency (Acc) (↑) | FPR (↓) | FNR (↓) |
> | :--- | :---: | :--- | :---: | :---: | :---: |
> | HardTests | 4o | AutoCode (4o) | 78.4% | 12.2% | **31.4%** |
> | **AutoCode (Ours)** | 4o | AutoCode (4o) | **79.5%** | **7.5%** | 34.2% |
> | | | | | | |
> | HardTests | o3 | AutoCode (o3) | 80.0% | 21.5% | 18.5% |
> | **AutoCode (Ours)** | o3 | AutoCode (o3) | **91.1%** | **3.7%** | **14.1%** |
>
> On the 4o model: AutoCode demonstrates higher consistency (79.5% vs 78.4%) and a lower False Positive Rate compared to the baseline. On the o3 model: The advantage of AutoCode is significant, with accuracy increasing to **91.1%** (compared to 80.0% for the baseline) and the False Positive Rate (FPR) dropping to an extremely low **3.7%**.
> Special Note on Experimental Setup:
> In the comparison experiments, the native Checkers generated by HardTests were not in the standard testlib.h format, meaning they could not be run directly on standard Online Judge systems. To ensure fairness and reproducibility, we used the standard-compliant Checkers generated by our AutoCode pipeline for all experiments listed above.
>
> **References**
>
> [1] Zhongmou He, Yee Man Choi, Kexun Zhang, Jiabao Ji, Junting Zhou, Dejia Xu, Ivan Bercovich, Aidan Zhang, and Lei Li. HardTests: Synthesizing High-Quality Test Cases for LLM Coding, 2025.
>
> [2] Zihan Wang, Siyao Liu, Yang Sun, Hongyan Li, Kai Shen. CodeContests+: High-Quality Test Case Generation for Competitive Programming, 2025.
>
> [3] Rongao Li, Jie Fu, Bo-Wen Zhang, Tao Huang, Zhihong Sun, Chen Lyu, Guang Liu, Zhi Jin, Ge Li. TACO: Topics in Algorithmic COde generation dataset, 2023.
>
> [4] Yujia Li, David Choi, Junyoung Chung, Nate Kushman, Julian Schrittwieser, Rémi Leblond, Tom Eccles, James Keeling, Felix Gimeno, Agustin Dal Lago, Thomas Hubert, Peter Choy, Cyprien de Masson d'Autume, Igor Babuschkin, Xinyun Chen, Po-Sen Huang, Johannes Welbl, Sven Gowal, Alexey Cherepanov, James Molloy, Daniel J. Mankowitz, Esme Sutherland Robson, Pushmeet Kohli, Nando de Freitas, Koray Kavukcuoglu, Oriol Vinyals. Competition-Level Code Generation with AlphaCode, 2022.
>
> ---
>
> **Q2. Reproducibility and Prompts**
>
> We fully understand your concern about reproducibility, as prompts are indeed a core part of our work. To address this, we have provided the full prompt templates for all key components in our anonymous repository (https://anonymous.4open.science/r/sfbgtqwerxzcwes-5BB0). We also commit to open-sourcing the entire codebase after the review process.
> Specifically, you can find the following template files in the repository:
>
> - test_generation/checker/checker_template.txt
>
> - test_generation/checker/checker_tests_template.txt
>
> - test_generation/generator/generator_template[1-3].txt
>
> - test_generation/validator/validator_template.txt
>
> - test_generation/validator/validator_tests_template.txt
>
> - test_generation/author/author_template.txt
>
> We believe these templates, which contain complete instructions and constraints, will fully support the reproducibility of our work.

---

> ### Author Response · Authors · 2025-11-21
>
> **Q3. Objective Verification of "Model-Training Usable"**
>
> We agree that objective metrics can supplement human evaluation. Therefore, we conducted a preliminary fine-tuning experiment to verify the "Model-Training Usable" criterion using Qwen3-8B-Thinking as the base model. We employed the Slime framework (https://github.com/THUDM/slime) to implement GRPO, using the 133 AutoCode-generated problems as training data and AutoCode-generated test cases as the reward signal. The model was trained for 5 epochs with a global batch size of 64, with a constant learning rate of 1e-6, and AdamW optimizer ($\beta_1 = 0.9, \beta_2 = 0.98$). During the rollout phase, we generated 8 samples per prompt with temperature = 0.8. We then compared the performance on LiveCodeBench before and after fine-tuning. The table below confirms that training on AutoCode data yields a clear accuracy improvement, offering preliminary validation for its effectiveness for model training.
>
> | Model | Metric | Base Model | Fine-tuned w/ AutoCode |
> | :--- | :--- | :---: | :---: |
> | **Qwen3-8B** | LiveCodeBench (Pass@1) | 46.2% | **48.5%** |
>
> We thank the reviewer for raising this point, and we commit to including results from larger-scale training in the final version of our paper to more rigorously demonstrate the utility of problems generated with our framework.
>
> ---
>
> **Q4. Human Evaluation Details**
>
> We apologize for not providing enough evaluation details in the initial submission. Here is the detailed information:
>
> **Expert Background:** We invited **5 human experts** to evaluate the LLM-generated problems. The group consisted of **2 Codeforces International Grandmasters** (reviewing high-difficulty problems) and **3 Codeforces Masters** (reviewing medium-to-low difficulty problems).
>
> **Evaluation Time:** Our experts were asked to comprehensively evaluate each problem's solvability, difficulty, novelty, quality, and the correctness of the LLM-provided solution. The average time spent reviewing each problem was approximately **30 minutes**.
>
> **Sample Size:** We generated and reviewed a total of 190 new problems, which is the data source for Figure 3. Within these, 133 problems were generated using our final prompt version, and these constitute the dataset for Figure 4.
> We will add these key numbers to the appendix of the paper to improve transparency and credibility.

---

> ### Author Response · Authors · 2025-11-21
>
> **Q5. Case Studies for Finding 2**
>
> We thank the reviewer for the constructive suggestion. To better support "Finding 2: LLMs tend to create new problems by combining existing algorithm frameworks and emphasizing knowledge/implementation rather than relying on novel insights," we provide three representative case studies below. These cases clearly show how the LLM increases difficulty by reorganizing and stacking known complex techniques.
>
> **Case 1: D2. Gapped Palindrome Pairs**
>
> Original Problem (D. Palindrome pairs): Count total non-overlapping palindrome substrings ($N \le 2000$). Solvable via $O(N^2)$ DP or enumeration.
>
> New Problem: For all possible gap lengths $g$, count non-overlapping palindrome pairs with exact gap $g$ ($N \le 200,000$).
>
> Reasoning: This is a typical "fusion" problem. To solve it efficiently, one must combine string processing (**Manacher's algorithm**) to identify palindromes, polynomial multiplication (**NTT/FFT**) to calculate counts for all gaps via convolution, and number theory (**CRT**) to ensure NTT precision. The difficulty increase relies entirely on stacking these advanced techniques and high implementation complexity.
>
> **Case 2: F2. Vlad and Too Many Errands**
>
> Original Problem (F. Vlad and Unfinished Business): Single query. Find the shortest path from $X$ to $Y$ visiting a set $A$. Relies on observing the minimal connected subgraph.
>
> New Problem: Multiple queries on the same large tree ($N, Q \le 200,000$), each with different sets $X, Y, A$.
>
> Reasoning: The core structural observation remains unchanged. However, to handle the constraints of large-scale multiple queries, the specific optimization technique of **"Virtual Trees"** must be introduced. The difficulty comes not from deeper graph theory insights, but from requiring the contestant to know and implement this specialized data structure (involving LCA, DFS order, etc.).
>
> **Case 3: D2. Zero Quantity Range Maximization**
>
> Original Problem (D. Zero Quantity Maximization): Given arrays $A$ and $B$, choose a global real number $d$ to maximize indices where $d \cdot A[i] + B[i] = 0$. Essentially finding the global mode of specific ratios.
>
> New Problem: Multiple queries. For a given range $[L, R]$, find the max zero count produced by a locally optimal $d$.
>
> Reasoning: The LLM converted the problem into a standard "Range Mode Query" problem. Under the constraints, the standard solution is **"Mo's Algorithm"** (offline square root decomposition). This merges the original simple algebraic observation with a mature but implementation-heavy optimization framework.
> We will add these detailed case studies to the appendix to provide sufficient evidence for Finding 2.
>
> ---
>
> **Q6. Problem Generation Process**
>
> We apologize for not describing the generation process in detail in the submission. We have provided the complete prompt template used for LLM problem generation in the anonymous repository (test_generation/author/author_template.txt).
> In this prompt, we assign the LLM the role of an "experienced problem setter" and set a series of strict quality criteria:
> Require the LLM to "modify at least one condition of the original problem."
> Explicitly require that the new problem "must be strictly harder than the original problem." We found this to be an effective way to guide the model toward substantive innovation.
> Require that the new problem "cannot be solved by heuristic methods."
> These specific instructions, combined with the "dual verification protocol" described in Section 5, constitute our core method for generating high-quality, verifiable new problems.
>
> ---
>
> **Q7. Sandbox System Development**
>
> Thank you for the detailed question. Our evaluation system is built on top of an open-source sandbox framework. At the lower level, we rely on Linux cgroup v2 to strictly monitor and limit resource usage, including runtime, memory consumption, and other system resources. The entire judging pipeline is encapsulated within a Docker container, where we enforce a unified compiler toolchain and fixed environment variables to ensure consistency across all evaluations. All experiments reported in the paper are conducted on the same server and within the same Docker environment, guaranteeing strict environment alignment. We treat the official verdicts as the ground truth, and thus all ablation studies are evaluated fairly under identical conditions.

---

> ### Comment · Reviewer_hPcp · 2025-11-27
>
> Thank you for your response. Some of my concerns have been addressed but I still I have several further suggestions. First, I believe using the generated problems for LLM RL training is a very good idea, and the authors have also added experimental validation on this matter. However, I think some details may still be missing, such as the training process (changes in rewards for the training and test sets, evolutions of key metrics like entropy and response length), as well as the version of LiveCodeBench not being specified. Given that RLVR is currently a crucial LLM training technique, many readers are likely to be interested in this aspect. Second, have the authors considered open-sourcing these generated problems? The cases provided by the authors seem promising, and synthetic data can also avoid many compliance risks, so open-sourcing these problems would be a significant contribution. Third, I noticed that the manuscript has not yet been revised, and I strongly suggest the authors incorporate the suggestions from the reviewers into the manuscript.

---

> > ### Author Response · Authors · 2025-11-28
> >
> > Thanks for your follow-up. We make the following three responses:
> >
> > ---
> >
> > **Q1: details about RL training and evaluation**
> >
> > We thank the reviewer for pointing this out the need for additional training details, which we agree is very important. Regarding the training dynamics, we observed that the reward consistently improved throughout the training process, the policy entropy slightly decreased, which is consistent with general observations in RL training. Regarding the LiveCodeBench version, we are using LiveCodeBench V5. Due to the time constraints of the rebuttal, the RL results are preliminary. We include the training curve of reward at https://anonymous.4open.science/r/sfbgtqwerxzcwes-5BB0/data/rollout-raw_reward.jpg and entropy at https://anonymous.4open.science/r/sfbgtqwerxzcwes-5BB0/data/train-entropy_loss.jpg. We will explore larger-scale training as well as a comprehensive analysis and presentation of training dynamics in future work.
> >
> > ---
> >
> > **Q2: open-sourcing AutoCode generated problems**
> >
> > We believe open-sourceing the generated problems by AutoCode will benefit the whole LLM for coding community. Therefore, we release the first version of the data at https://anonymous.4open.science/r/sfbgtqwerxzcwes-5BB0/data/synthetic_problems_133.jsonl, and will expand and make it public after the review process.
> >
> > ---
> >
> > **Q3: revised manuscript**
> >
> > Thanks for pointing out. We've updated the manuscript based on all reviewers' comments. We add four section in Appendix ranging from page 25 to 34, covering prompt templates, early results of RL on AutoCode data, fair comparison with Hardtests, and human evaluation process and requirements. We also slightly revise the related work section.

---

### Official Review · Reviewer_2TeP · 2025-11-03

**Soundness:** 3
**Presentation:** 3
**Contribution:** 2
**Rating:** 6
**Confidence:** 3

**Summary:**

Auto-code is about being able to generate new competitive programming problems. This requires being able to generate new problem descriptions from scratch or by modifying existing coding problem descriptions. And it requires providing reference solutions to the new problems to show they are solvable and well formed.  And to do that requires being able to generate high quality test cases and be able to verify if a code solution can pass all the test cases, and verify the code solution is solving the new programming problem description.

This paper describes a pipeline of many agents that work together to accomplish that task, and benchmarks some of the components in the pipeline against other approaches previously tried.

**Strengths:**

The paper describes a system of agents that produce new competitive programming problems and solutions that appear to be of high quality when judged by human experts.  The paper does nicely describe the rigorous validation and verification required to vet out proposed problems and their proposed solutions as correct or incorrect, by ensuring a reliable and broad set of test cases is produced. The paper does a number of comparisons that shows the quality of the system, like how well it is able to generate test cases compared to other test case generation systems.

**Weaknesses:**

The description of the system in the paper is well written and extensive, but I do worry that without a repo of the source code accessible to researchers the ability to leverage this work and build upon it will be very difficult.

The ability to produce new competitive programming problems that humans judge as good is useful on it's own, but if those new problems and correct solutions are able to be leveraged to further finetune the LLM and improve it's ability to solve competitive programming problems, that would be strong indicator that the problems being generated are useful in an interesting way.

**Questions:**

Are you planning to release the source code for the system described in the paper?

Are you able to leverage the generated programming problems and solutions to finetune an LLM and show it leads to the LLM being a better problem solver?

---

> ### Author Response · Authors · 2025-11-21
>
> We thank the reviewer for the positive assessment of our system's quality and the rigorous validation process. We address your questions regarding code release and fine-tuning utility below.
>
> ---
>
> **Q1. Source Code Release**
>
> Yes, we fully intend to open-source our system. We deeply understand the importance of reproducibility for the community. Currently, as noted in our other responses, we have provided an anonymous repository containing the core code and all prompt templates here: https://anonymous.4open.science/r/sfbgtqwerxzcwes-5BB0. We commit to publicly releasing the full project in the future.
>
> ---
>
> **Q2. Utility for Fine-tuning**
>
> To substantiate the utility of problems generated with our AutoCode framework, we conducted a fine-tuning experiment using Qwen3-8B-Thinking as the base model. We employed the Slime framework (https://github.com/THUDM/slime) to implement GRPO, using the 133 AutoCode-generated problems as training data and AutoCode-generated test cases as the reward signal. The model was trained for 5 epochs with a global batch size of 64, with a constant learning rate of 1e-6, and AdamW optimizer ($\beta_1 = 0.9, \beta_2 = 0.98$). During the rollout phase, we generated 8 samples per prompt with temperature = 0.8. We then compared the performance on LiveCodeBench before and after fine-tuning. The table below confirms that training on AutoCode data yields a clear accuracy improvement, offering preliminary validation for its effectiveness for model training.
>
> | Model | Metric | Base Model | Fine-tuned w/ AutoCode |
> | :--- | :--- | :---: | :---: |
> | **Qwen3-8B** | LiveCodeBench (Pass@1) | 46.2% | **48.5%** |
>
> We thank the reviewer for raising this question, and we commit to including results from larger-scale training in the final version of our paper to more rigorously demonstrate the utility of problems generated with our framework.

---

### Author Response · Authors · 2025-11-29

We thank the reviewers for all their comments, which helped us improve the paper a lot. We appreciate that the reviewers agreed on the quality of our writing, the soundness of our framework, and the professionalism of the human expert evaluation. During the rebuttal process, we updated the manuscript based on all reviewers’ comments. We added four sections in the Appendix (pages 25–34), covering prompt templates, early RL results on the AutoCode data, fair comparison with HardTests, and the human evaluation process and requirements. We also made small edits to the related work section. We especially thank Reviewer gTC1 for accepting our response and increasing the score from 6 to 8.

---

### Meta-Review · Area_Chair_cN7p · 2026-01-08

**Summary:**

This submission introduces AutoCode, a framework that uses LLMs to act as competitive-programming problem setters rather than just solvers. The paper received reviews from three reviewers, resulting in divergent scores.  All reviewers recognized the novelty of the "LLM-as-Problem-Setter" paradigm and the depth of human evaluation. The core of the main concerns related to a lack of reproducibility and the fairness of baselines compared to SOTA.

The rebuttal successfully addressed these concerns by releasing an anonymous repository containing full prompt templates and core code, adding a comparison against the SOTA baseline. These revisions led the reviewer (gTC1) to explicitly raise their score from 6 to 8. And one reviewer (hPcp) shows a positive attitude. Another reviewer (2TeP), however, did not participate in the discussion, but their core concerns, such as the reproducibility and further RL experiments, were well addressed.

Therefore, AC decides to recommend acceptance at this stage.

**Reviewer Concerns:**

Addressed:
1. Reproducibility: The authors provided an anonymous repository containing full prompt templates and core code.
2. Fair Baselines: A direct, controlled comparison against the HardTests baseline was added.
3. More Benchmark Results: The authors provided more results showing that AutoCode-generated problems improve performance on LiveCodeBench.

Unresolved:
1. More RL Analysis and Bigger Model: Detailed training dynamics and larger-scale experiments are still needed for the final version.

**Reviewer Scores:**

Reviewer 2TeP likely maintained their score of 6, since the new RL experiments substantiated their initially positive assessment regarding the utility of the generated problems.

Reviewer hPcp could have changed its score from 2 to 4, since the inclusion of fair baseline comparisons and preliminary RL training results partially addressed their concerns about objective verification.

Reviewer gTC1 has indicated that their score was raised from 6 to 8, after the authors clarified implementation details and provided the missing prompts to ensure reproducibility.

---

### Decision · Program_Chairs · 2026-01-26

Accept (Poster)